# CircAnks1a in the spinal cord regulates hypersensitivity in a rodent model of neuropathic pain

Su-Bo Zhang[1,2,6], Su-Yan Lin[2,3,6], Meng Liu[2,3,6], Cui-Cui Liu [1,6], Huan-Huan Ding[2,3], Yang Sun[4], Chao Ma[1], Rui-Xian Guo[2], You-You Lv[3], Shao-Ling Wu[1], Ting Xu[2] & Wen-Jun Xin [2,3,5]

Circular RNAs are non-coding RNAs, and are enriched in the CNS. Dorsal horn neurons of the spinal cord contribute to pain-like hypersensitivity after nerve injury in rodents. Here we show that spinal nerve ligation is associated with an increase in expression of circAnks1a in dorsal horn neurons, in both the cytoplasm and the nucleus. Downregulation of circAnks1a by siRNA attenuates pain-like behaviour induced by nerve injury. In the cytoplasm, we show that circAnks1a promotes the interaction between transcription factor YBX1 and transportin-1, thus facilitating the nucleus translocation of YBX1. In the nucleus, circAnks1a binds directly to the *Vegfb* promoter, increases YBX1 recruitment to the *Vegfb* promoter, thereby facilitating transcription. Furthermore, cytoplasmic circAnks1a acts as a miRNA sponge in miR-324-3p-mediated posttranscriptional regulation of VEGFB expression. The upregulation of VEGFB contributes to increased excitability of dorsal horn neurons and pain behaviour induced by nerve injury. We propose that circAnks1a and VEGFB are regulators of neuropathic pain.

[1] Guangdong Provincial Key Laboratory of Malignant Tumor Epigenetics and Gene Regulation, Department of Rehabilitation Medicine, Sun Yat-Sen Memorial Hospital, Sun Yat-Sen University, Guangzhou 510120, China. [2] Guangdong Province Key Laboratory of Brain Function and Disease, Department of Physiology, Zhongshan School of Medicine, Sun Yat-Sen University, Guangzhou 510080, China. [3] Neuroscience Program, Zhongshan School of Medicine, the Fifth Affiliated Hospital of Sun Yat-Sen University, Sun Yat-Sen University, Guangzhou, China. [4] Department of Rehabilitation Medicine, the Second Affiliated Hospital of Xi'an Medical University, Xi'an 710038, China. [5] Center for Brain Science and Brain-Inspired Intelligence, Guangdong-Hong Kong-Macao Greater Bay Area, Guangzhou, China. [6]These authors contributed equally: Su-Bo Zhang, Su-Yan Lin, Meng Liu, Cui-Cui Liu. Correspondence and requests for materials should be addressed to C.M. (email: machao@mail.sysu.edu.cn) or to T.X. (email: xuting8@mail.sysu.edu.cn) or to W.-J.X. (email: xinwj@mail.sysu.edu.cn)

Many signaling cascades and molecules, such as the neuroinflammatory response, purine receptors, endogenous opioids and neurotrophins, are involved in the development of neuropathic pain[1–4]; however, the molecular mechanism underlying neuropathic pain remains unclear. Circular RNAs (circRNAs) are endogenous noncoding RNAs that form covalently closed loops[5,6]; their sequences are conserved among species and they have higher stability than linear mRNAs[7,8]. Most circRNAs are detected at a higher abundance in mammalian brain than other surveyed tissues[9,10], suggesting that they may play a vital role in pathological processes and may serve as biomarkers for some neurological disorders[11]. CircRNAs have been shown to act as competing endogenous RNAs that sponge miRNAs by complementary base pairing and regulate the translation of target mRNAs. For example, ciRS-7 functions as a miRNA sponge to suppress miR-7 activity[12]. In addition, there is evidence that some circRNAs may form RNA–protein complexes with RNA-binding proteins and regulate their activities[13,14]. Although recent studies showed that nerve injury alters circRNA expression in rat spinal dorsal horn[15,16], whether and how circRNA contributes to neuropathic pain has not been reported.

VEGFB, as the least well-characterized member of the VEGF family, is widely expressed in various tissues, including muscle, brain, and fat[17,18]. Due to its high sequence homology to VEGFA and similar receptor binding pattern, VEGFB was originally thought to be an angiogenic factor[19] until a series of studies demonstrated its involvement in various physiological and pathological scenarios[20,21]. For example, VEGFB mRNAs were significantly increased within temporal white matter following radiation-induced brain injury[22], and VEGFB was also found to modulate central nervous system inflammation and thereby worsen encephalomyelitis[23]. However, whether and how VEGFB participates in neuropathic pain still remains elusive.

In this study, we show for the first time that a spinal cord-specific circRNA, circAnks1a, promoted the translocation of the transcription factor YBX1 into the nucleus. Furthermore, nuclear circAnks1a facilitated the interaction of YBX1 and the *Vegfb* promoter via specific RNA–DNA interaction following nerve injury. Cytoplasmic circAnks1a also served as a miRNA sponge for miR-324-3p and enhanced VEGFB mRNA translation. The upregulation of VEGFB excited the dorsal horn neurons and contributed to pain behavior induced by nerve injury. These findings reveal a novel mechanism and identify specific targets for the treatment of neuropathic pain.

## Results

**Sequencing of circRNAs in spinal dorsal horn**. To generate a circRNA profiling database, we collected spinal dorsal horn tissue from rats following spinal nerve ligation (SNL) and examined the expression of circRNA using RNA-seq analysis of ribosomal RNA-depleted and RNase R-treated RNA. Sham (Day 14) and SNL-treated tissues (Days 7 and 14) were sequenced on an Illumina HiSeq platform, yielding an average of 70 million reads, which were mapped onto the rat genome (Rnor_6.0) using TopHat2 (Supplementary Table 1)[24]. We detected 61,833 distinct circRNAs based on the criterion of at least one unique back-spliced junction read. The reads per million mapped reads (RPM) of 12,849 of these circRNAs was greater than 0.1. We annotated these circRNAs using the RefSeq database and found that 65.02% of the circRNAs consisted of protein-coding exons (Fig. 1a). Information on the length of the detected circRNAs (minimum, maximum, and median) is provided in Fig. 1b and Supplementary Table 2.

To identify the critical circRNAs in the pathogenesis of neuropathic pain, we compared the circRNAs in the sham group

and the SNL-treatment group. The DESeq analysis showed that 21 circRNAs were significantly dysregulated with >2.5-fold change on days 7 and 14 following SNL (Fig. 1c). Using circRNA-specific divergent primers, quantitative polymerase chain reaction (qPCR) analysis was performed to examine the changes in the levels of 21 circRNAs (Supplementary Table 3). We found that circ: chr20:7561057-7573740 (circAnks1a) showed the most significant increase on days 3, 7, 10, and 14 after SNL treatment (Fig. 1d, Supplementary Table 3).

**Characterization of circAnks1a in spinal dorsal horn**. In silico analysis predicted that circAnks1a was derived from exon 5 to exon 11 of the *Anks1a* gene (988 bp) (Fig. 2a). Sanger sequencing verified this prediction (Fig. 2b). To confirm the circular characteristics of circAnks1a, random hexamer or oligo (dT)$_{18}$ primers were used in reverse transcription experiments. Compared with random hexamer primers, the relative expression of circAnks1a, but not Anks1a mRNA (mAnks1a), was barely detected when the primers were replaced by oligo (dT)$_{18}$ (Fig. 2c). Furthermore, we found that the dorsal horn circAnks1a was resistant to digestion by RNase R (Fig. 2d). We also validated the specificity of circAnks1a in dorsal horn (L4–L5) using northern blotting. The results showed that circAnks1a could be detected using a circAnks1a probe in samples that had undergone RNase R treatment, whereas the mAnks1a probe only detected mAnks1a in samples that had not been treated with RNase R (Fig. 2e). The results confirmed the specificity of cyclization of circAnks1a. Next, we intrathecally (i.t.) injected actinomycin D to inhibit transcription and measured the content of circAnks1a and mAnks1a in the spinal dorsal horn. CircAnks1a showed slower degradation and a longer half-life (more than 24 h) than mAnks1a (Fig. 2f). These results demonstrate that circAnks1a is a stable circular transcript. In addition, we measured the abundance of circAnks1a in various tissues following SNL. The results showed that circAnks1a was predominantly expressed in spinal dorsal horn in normal rats and that SNL significantly increased the expression of circAnks1a only in the dorsal horn, suggesting that circAnks1a is a well-expressed spinal/pain-specific (tissue/status-specific) circRNA (Fig. 2g). We measured the expression of circAnks1a at different levels of the spinal cord from T13 to S3; the highest expression was found in laminae I-III of L4–L6 (Supplementary Fig. 1a). Fluorescence in situ hybridization (FISH) and qPCR assays further demonstrated that circAnks1a was expressed in the nucleus and cytoplasm of dorsal horn neurons (Fig. 2h, i). As in other studies[5], conservative analysis of the full-length sequence in 20 different species was conducted using Phylop, and circAnks1a was found to contain 66 conservative blocks (http://compgen.cshl.edu/phast/). In addition, the full-length sequence of circAnks1a was blasted against the human circRNAs in circBase, and circAnks1a was found to have high homology with three human circRNAs that originate from *Anks1a*: hsa_circ_0076077, hsa_circ_0076079, and hsa_circ_0076081 (Supplementary Fig. 1b). These results suggest that the biogenesis of circAnks1a is conserved in mammals, while its function has not been reported.

**Increased circAnks1a-mediated pain-like hypersensitivity**. To verify that circAnks1a is involved in chronic pain, electrophysiological alterations, and changes in pain behavior in the treated animals were examined. We designed two small interfering RNAs (siRNAs); one siRNA targeted the backsplice sequence (circAnks1a siRNA) and was designed to silence the expression of circAnks1a, and the other targeted the linear transcript (mAnks1a siRNA) and was designed to knockdown mAnks1a (Supplementary Fig. 1c). As expected, intrathecal

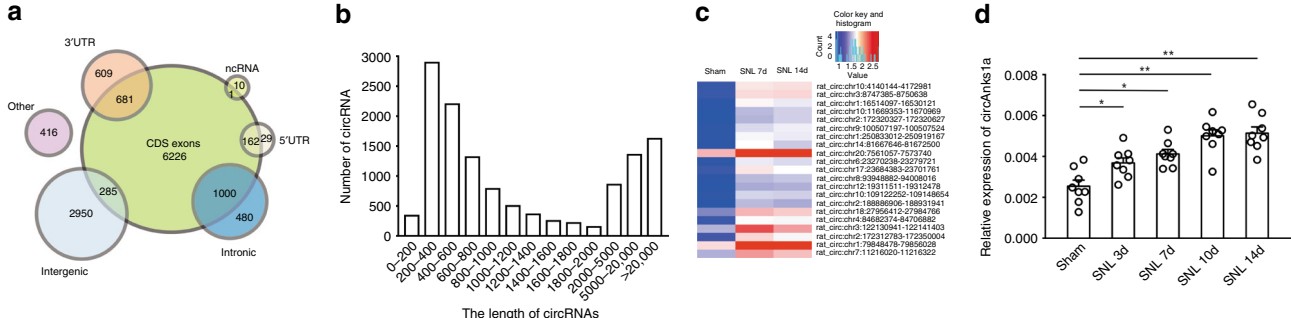

**Fig. 1** Profiling the circRNAs in spinal dorsal horn of rats. **a** Genomic origin of circRNAs. **b** The length distribution of circRNAs. **c** Heat map of the top 21 increased circRNAs on days 7 and 14 following SNL treatment. **d** Relative expression level of circAnks1a in spinal dorsal horn at various time points after SNL treatment ($^*P < 0.05$, $^{**}P < 0.01$ vs. the sham group, two-tailed one-way ANOVA, $n = 8$). The data are presented as the mean ± s.e.m. Source data are available as a Source Data file

injection of circAnks1a siRNA (1 nmol 10 μl$^{-1}$) on 5 consecutive days specifically knocked down the expression of circAnks1a, whereas intrathecal injection of mAnks1a siRNA specifically knocked down the expression of mAnks1a on day 14 following SNL and in naïve rats (Fig. 3a and Supplementary Fig. 1d). Electrophysiological studies showed that the frequency and amplitude of miniature excitatory postsynaptic currents (mEPSCs), depolarization-induced neuronal firing, and EPSP-spike coupling were significantly increased in NK1R-positive neurons (in which immunoreactivity was colocalized with circAnks1a-positive cells) in the spinal cord slices on days 7 and 14 after nerve injury (Fig. 3b and Supplementary Fig. 2). Furthermore, in vivo experiments showed that the number of C-fiber–evoked action potentials increased after SNL (Fig. 3c). Inhibition of circAnks1a by injection of circAnks1a siRNA ameliorated the increase in the amplitude and frequency of mEPSCs, the positive shift in the resting membrane potential, the increase in the number of C-fiber–evoked action potentials, the depolarization-induced neuronal firing and the EPSP-spike coupling that were otherwise observed on days 7 and 14 after SNL (Fig. 3b, c and Supplementary Fig. 2), whereas inhibition of mAnks1a did not attenuate these neuronal activities on day 14 following nerve injury (Fig. 3b, c and Supplementary Fig. 2). A subsequent behavioral test indicated that knockdown of circAnks1a, but not knockdown of mAnks1a, significantly elevated the withdrawal threshold (Fig. 3d) and the withdrawal latency (Fig. 3e) in the SNL-treated rats. Importantly, we also noted that circAnks1a siRNA or mAnks1a siRNA injection per se did not change the behavioral sensitivity in the sham rats (Fig. 3d, e). To further test whether circAnks1a in the dorsal horn contributes to chronic pain, 600 nl of recombinant AAV-hSyn-circAnks1a-nEF1α-EGFP was intraspinally injected into the L4–L6 spinal cord to overexpress circAnks1a. Twenty-one days after virus injection, the presence of marked green fluorescence and increased circAnks1a levels in the dorsal horn suggested a high efficiency of transfection (Supplementary Fig. 3a, b). Importantly, the number of EPSP-spike couplings (Supplementary Fig. 3c) and the amplitude and frequency of mEPSCs (Fig. 3f) increased, and the withdrawal threshold (Fig. 3g) and the withdrawal latency (Fig. 3h) declined in AAV-circAnks1a-EGFP-injected rats relative to AAV-EGFP-injected rats. These results suggest that the spinal cord-specific and conserved circAnks1a contributes to central sensitization and behavioral hypersensitivity and that it may represent a novel target for the treatment of chronic pain.

**VEGFB contributed to neuron excitation and neuropathic pain.** To elucidate the molecular mechanism through which circAnks1a regulates neuropathic pain, we measured the gene expression in the dorsal horn of AAV-injected naïve rats using whole genome expression microarrays. Compared with AAV-EGFP-injected rats, the expression of 29 transcripts was upregulated by more than twofold in AAV-circAnks1a-EGFP-injected rats (Fig. 4a). Among these transcripts, VEGFB mRNA showed the maximum increase with a $q < 0.05$ (Fig. 4b). Furthermore, in the SNL model, VEGFB mRNA and protein were also significantly increased on days 3, 7, 10, and 14 compared with the sham group (Fig. 4c, d). In addition, VEGFB was expressed only in neurons (NeuN), but not in astrocytes (GFAP) or microglia (Iba-1) in the dorsal horn (Fig. 4e). To determine whether the upregulated VEGFB contributes to central sensitization and nociceptive transmission, we intrathecally injected VEGFB siRNA and examined the resulting alterations in electrophysiological characteristics and pain behavior. The decreased VEGFB mRNA and protein expression following injection of VEGFB siRNA suggested the efficacy of transfection (Supplementary Fig. 4a, b). Compared with scramble siRNA, intrathecal injection of VEGFB siRNA (1 nmol 10 μl$^{-1}$ for 5 consecutive days) significantly ameliorated the increase in mEPSC amplitude and frequency (Fig. 4f). Importantly, the withdrawal threshold and the withdrawal latency in SNL rats, but not those in sham rats, increased significantly following injection of VEGFB siRNA (Fig. 4g, h). Next, recombinant AAV encoding Cre and mCherry (AAV-Cre-mCherry) was intraspinally injected into VEGFB$^{flox/flox}$ mice. Twenty-one days after virus injection, marked red fluorescence and decreased expression of VEGFB mRNA suggested a high efficiency of transfection (Supplementary Fig. 4c, d). Importantly, mechanical allodynia (Fig. 4i) and thermal hyperalgesia (Fig. 4j) were greatly attenuated in AAV-Cre-mCherry injected VEGFB$^{flox/flox}$ mice relative to AAV-mCherry-injected VEGFB$^{flox/flox}$ mice after SNL. Finally, we overexpressed VEGFB by injecting AAV-VEGFB-EGFP into the L4–L6 dorsal horn (Supplementary Fig. 4e, f). Compared with AAV-EGFP, overexpression of VEGFB significantly decreased the withdrawal threshold and the withdrawal latency in normal rats (Fig. 4k, l). These results suggest that VEGFB plays a critical role in the development of neuropathic pain induced by nerve injury.

**CircAnks1a-regulated VEGFB upregulation in neuropathic pain.** We further examined whether VEGFB upregulation was mediated by circAnks1a in SNL-induced chronic pain. The results showed that circAnks1a colocalized with VEGFB (Fig. 5a), and intrathecal injection of circAnks1a siRNA reduced the upregulation of VEGFB mRNA (Fig. 5b) and protein (Fig. 5c) in the dorsal horn on day 14 following SNL. Furthermore, circAnks1a overexpression by intraspinal injection of the recombinant AAV-circAnks1a-EGFP significantly increased the mRNA (Fig. 5d) and

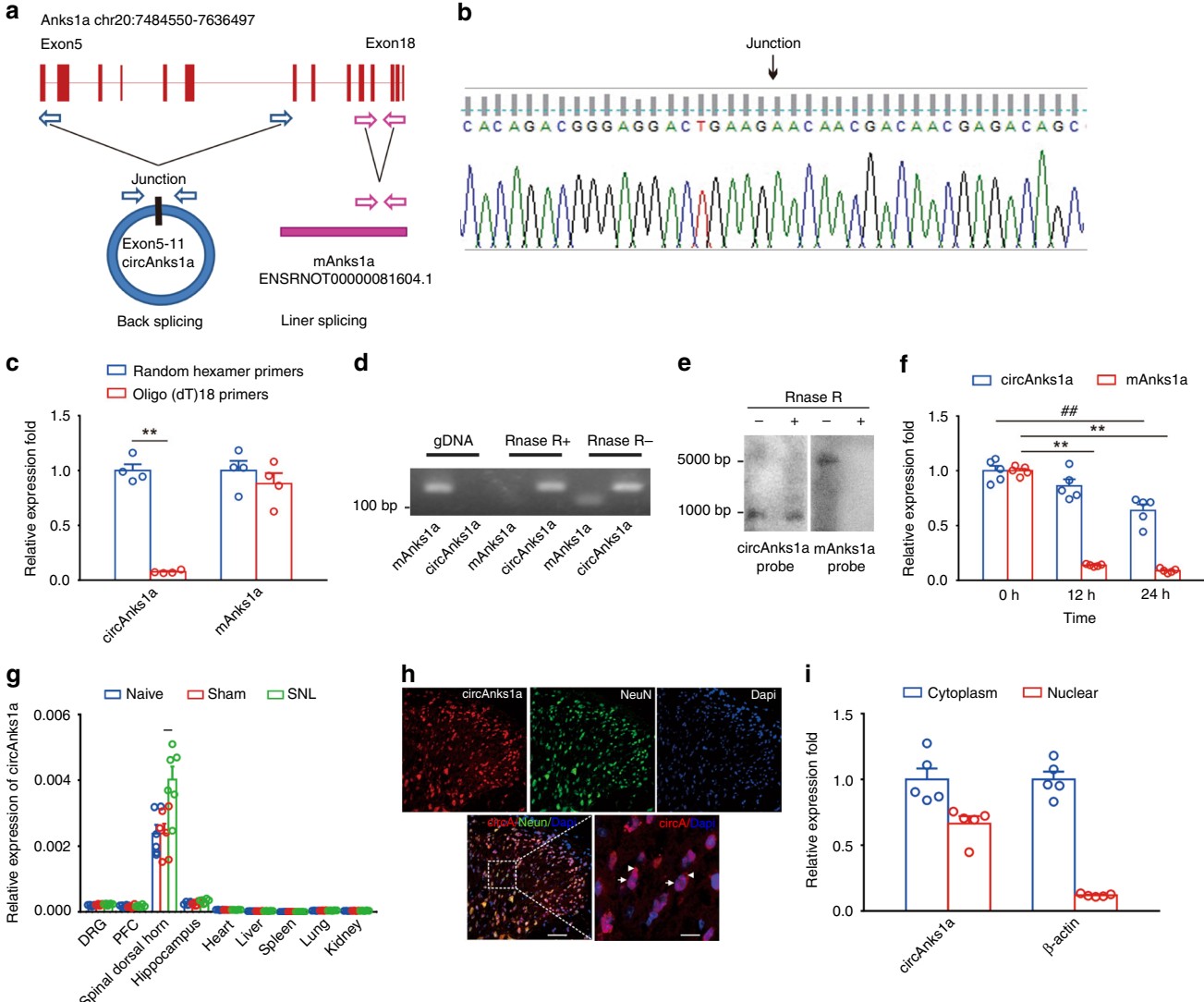

**Fig. 2** Identification of circAnks1a. **a** Scheme illustrating the production of circAnks1a. CircAnks1a was formed by back splicing from exon 5 to exon 11 of the *Anks1a* gene. **b** The expression of circAnks1a was validated by Sanger sequencing. The arrow indicates the junction site of circAnks1a. **c** Random hexamer or oligo(dT)$_{18}$ primers were used in the reverse transcription experiments. The RNA levels were analyzed by qPCR and normalized to the value obtained using random hexamer primers (**$P < 0.01$ vs. random hexamer primers group, two-tailed two-sample $t$ tests, $n = 4$). **d** PCR showed that dorsal horn circAnks1a, but not mAnks1a, was resistant to digestion by RNase-R ($n = 3$). **e** Northern blotting showed that circAnks1a was detected by circAnks1a probes after RNase R treatment, while mAnks1a was not ($n = 3$). **f** The relative RNA levels of circAnks1a and mAnks1a in the spinal dorsal horn were analyzed by qPCR at various time points after treatment with actinomycin D (i.t.) (**$P < 0.01$ vs. the mAnks1a group at 0 h, ##$P < 0.05$ vs. the circAnks1a group at 0 h, two-tailed one-way ANOVA, $n = 5$). **g** The expression of circAnks1a in nine different tissues was examined in naïve, sham and SNL-treated rats (**$P < 0.01$ vs. the sham group, two-tailed one-way ANOVA, $n = 6$). **h** FISH showed that circAnks1a was expressed in the nucleus (arrow) and cytoplasm (arrowhead) of spinal dorsal horn neurons. Left scale bar, 50 μm; right scale bar, 10 μm ($n = 3$). **i** PCR results obtained after nucleus and cytoplasm separation showed circAnks1a was expressed in both the nucleus and the cytoplasm ($n = 5$). The data are presented as the mean ± s.e.m. Source data is available as a Source Data file

protein (Fig. 5e) levels of VEGFB. These results suggest that upregulation of VEGFB in the dorsal horn after SNL is dependent on circAnks1a expression.

**CircAnks1a promoted the nuclear translocation of YBX1.** To elucidate the molecular mechanism by which circAnks1a regulates VEGFB expression in the dorsal horn following SNL, we performed an RNA pulldown assay using exogenous linearized circAnks1a to search for potential circAnks1a-associated proteins. Mass spectrometry (Supplementary Fig. 5) and western blotting (Fig. 6a) revealed an obvious interaction between linearized

circAnks1a and YBX1 protein on day 14 following SNL. To examine whether endogenous circAnks1a also bound to YBX1 protein following SNL, we performed the RNA pulldown assay using a circAnks1a probe. Compared with the sham group, YBX1 was significantly immunoprecipitated by endogenous circAnks1a on day 14 following SNL (Fig. 6b). In addition, RNA-binding protein immunoprecipitation (RIP) analysis showed that the level of circAnks1a precipitated by the YBX1 antibody was significantly increased on day 14 following SNL compared with the sham group in dorsal horn tissues (Fig. 6c). Bioinformatics analysis (ATtRACT, RBPmap) showed that two potential YBX1 motifs were present in the circAnks1a (Fig. 6d). Furthermore, the EMSA

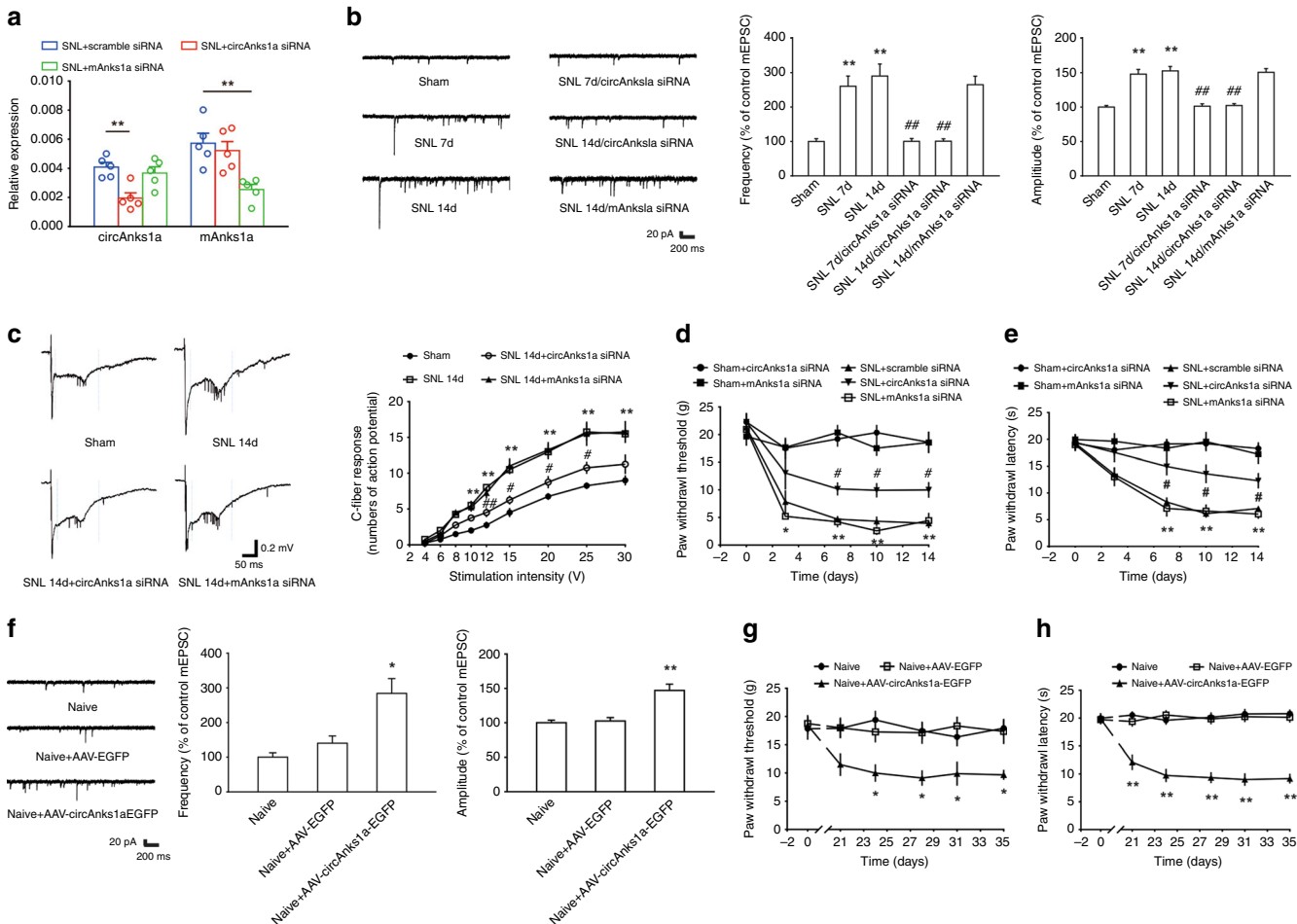

**Fig. 3** Upregulation of circAnks1a-mediated neuropathic pain following SNL. **a** Application of circAnks1a siRNA but not mAnks1a siRNA (1 nmol 10 μl$^{-1}$ i.t. for consecutive 5 days) knocked down the expression of circAnks1a on day 14 following SNL ($^{**}P < 0.01$ vs. the corresponding scramble group, two-tailed one-way ANOVA, $n = 5$). **b** CircAnks1a siRNA treatment ameliorated the increase in the amplitude and frequency of mEPSCs on days 7 and 14 induced by SNL ($^{**}P < 0.01$ vs. the sham group, $^{\#\#}P < 0.01$ vs. the corresponding SNL group, two-tailed one-way ANOVA for amplitude, two-tailed permutation tests for frequency, $n = 22$). **c** The number of C-fiber–evoked action potentials increased following SNL ($^{**}P < 0.01$ vs. the sham group, $^{\#}P < 0.05$, $^{\#\#}P < 0.01$ vs. the corresponding SNL group, two-tailed one-way ANOVA, $n = 4$). **d** Intrathecal injection of circAnks1a siRNA attenuated the mechanical allodynia induced by SNL ($^{*}P < 0.05$, $^{**}P < 0.01$ vs. the sham + circAnks1a siRNA group, $^{\#}P < 0.05$ vs. the SNL + scramble siRNA group, two-tailed permutation tests, $n = 10$). **e** The effect of circAnks1a siRNA on withdrawal latency ($^{**}P < 0.01$ vs. the sham + circAnks1a siRNA group, $^{\#}P < 0.05$ vs. the SNL + scramble siRNA group, two-tailed permutation tests, $n = 12$). **f** Intraspinal injection of recombinant AAV-circAnks1a-EGFP increased the amplitude and frequency of mEPSCs ($^{*}P < 0.05$, $^{**}P < 0.01$ vs. the AAV-EGFP group, two-tailed one-way ANOVA for amplitude, two-tailed permutation tests for frequency, $n = 18$). **g** Recombinant AAV-circAnks1a-EGFP significantly reduced the withdrawal threshold 21 days after injection ($^{*}P < 0.05$ vs. the AAV-EGFP group, two-tailed permutation tests, $n = 12$). **h** The effect of AAV-circAnks1a-EGFP on withdrawal latency ($^{**}P < 0.01$ vs. the AAV-EGFP, two-tailed permutation tests, $n = 12$). The data are presented as the mean ± s.e.m. Source data are available as a Source Data file

experiment showed that recombinant YBX1 protein obviously bound to the circAnks1a site-1 (YBX1 motif-1), but not to site-2 (YBX1 motif-2), and preincubation of unlabeled probe, but not mut-unlabeled probe, prevented the binding of YBX1 to circAnks1a site-1 (Supplementary Fig. 6a). These results suggest that the exact site of circAnks1a to which YBX1 binds is the UCCAGCAA sequence. High resolution images obtained by structured illumination microscopy (SIM) showed that circAnks1a bound to YBX1 in both the cytoplasm and the nucleus (Fig. 6e). Transcription factors such as YBX1 often execute their functions after translocating into the nucleus. Hence, we separated the nucleus and cytoplasm of spinal dorsal horn tissues, and measured the levels of nuclear YBX1 following SNL. The results showed that the amount of nuclear YBX1, but not the amount of total YBX1, increased significantly on days 7 and 14 following SNL (Fig. 6f and Supplementary Fig. 6b), and this increase displayed a time course similar to that of the increase in circAnks1a.

We further investigated whether circAnks1a participated in the nuclear translocation of YBX1. Intrathecal injection of circAnks1a siRNA inhibited nuclear YBX1 accumulation on day 14 after SNL (Fig. 6g), and intraspinal injection of AAV-circAnks1a-EGFP markedly enhanced the level of nuclear YBX1 (but not that of total YBX1) in naïve rats compared with naïve rats injected with AAV-EGFP (Fig. 6h and Supplementary Fig. 6d). It is well-known that transportin-1 plays an important role in cytoplasmic–nuclear transport[25], and that it mediates YBX1 nuclear translocation in HeLa cells[26]. It is unclear whether the increased YBX1 found in the nucleus following SNL is associated with transportin-1. Significantly increased amounts of YBX1 were found in the immunocomplexes precipitated by transportin-1 from dorsal horn lysates on day 14 after SNL (Fig. 6i), suggesting an enhanced interaction between YBX1 and transportin-1. Importantly, intrathecal injection of circAnks1a siRNA significantly attenuated the interaction between YBX1 and transportin-1 in the SNL

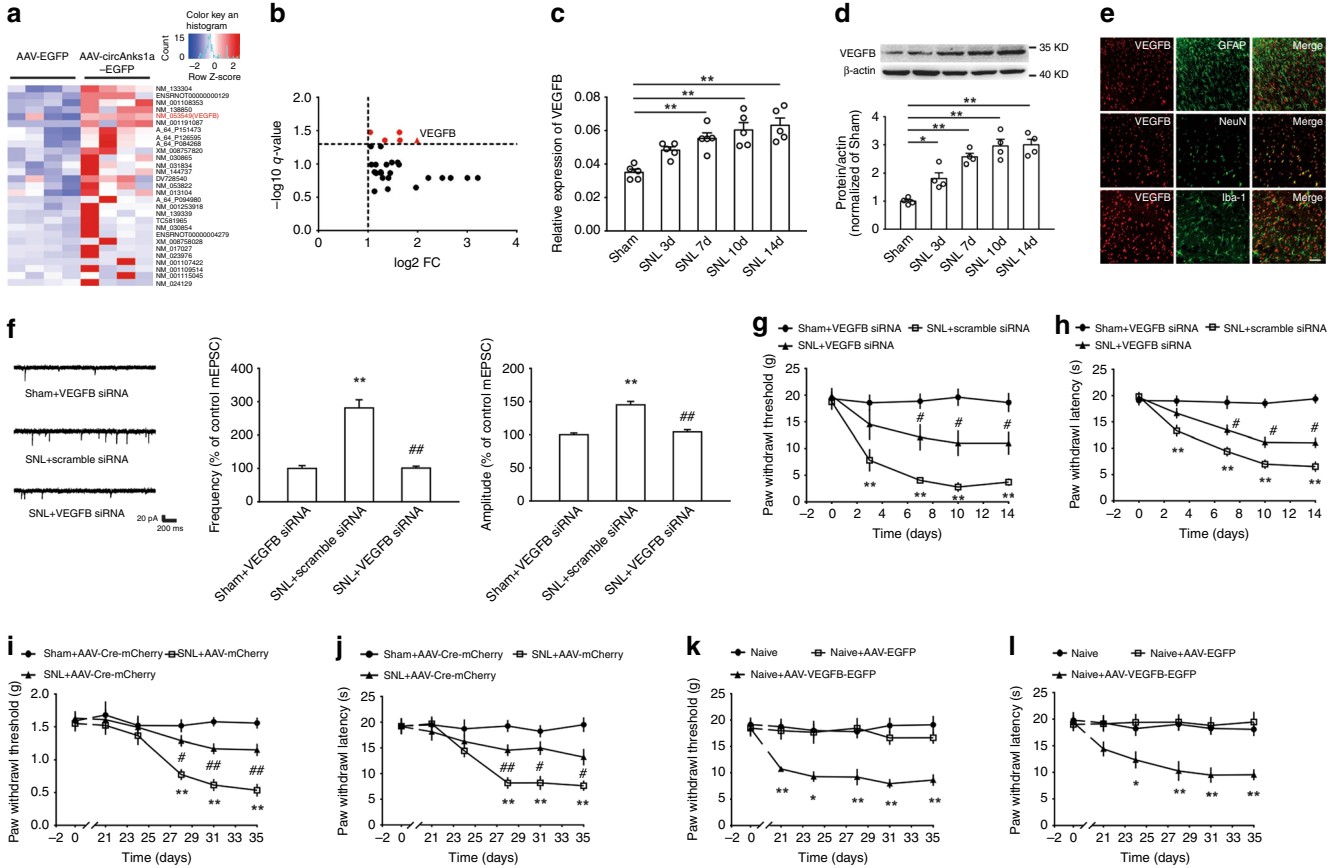

**Fig. 4** VEGFB in the spinal dorsal horn contributed to neuropathic pain following SNL. **a** Heat map showed the significantly upregulated mRNAs in the dorsal horn of AAV-circAnks1a-EGFP-injected rats ($n = 4$). **b** Volcano map showing that VEGFB has the maximum increased fold with $q < 0.05$ ($n = 4$). **c, d** The mRNA and protein levels of VEGFB in the dorsal horn of rats were measured at various time points following SNL (*$P < 0.05$, **$P < 0.01$ vs. the sham group, two-tailed one-way ANOVA, $n = 5$ for (**c**) and $n = 4$ for (**d**)). **e** Immunofluorescence staining of VEGFB (red) colocalized with NeuN (neuron marker, green) but not with GFAP (astrocyte marker, green) or Iba-1 (microglial marker, green). Scale bar, 50 μm ($n = 3$). **f** VEGFB siRNA treatment (1 nmol 10 μl⁻¹, i.t.) ameliorated the increase in the amplitude and frequency of mEPSCs on day 14 following SNL (**$P < 0.01$ vs. the sham group, ##$P < 0.01$ vs. the corresponding scramble group, two-tailed one-way ANOVA for amplitude, two-tailed permutation tests for frequency, $n = 24$). **g** Intrathecal injection of VEGFB siRNA attenuated the mechanical allodynia induced by SNL (**$P < 0.01$ vs. the sham group, #$P < 0.05$ vs. the corresponding scramble group, two-tailed permutation tests, $n = 12$). **h** The effect of VEGFB siRNA on withdrawal latency (**$P < 0.01$ vs. the sham group, #$P < 0.05$ vs. the corresponding scramble group, two-tailed permutation tests, $n = 12$). **i, j** Local knockdown of VEGFB by intraspinal injection of AAV-Cre-mCherry into VEGFB^flox/flox mice significantly ameliorated mechanical allodynia and thermal hyperalgesia (**$P < 0.01$ vs. the sham group, #$P < 0.05$, ##$P < 0.01$ vs. the corresponding AAV-mCherry group, two-tailed permutation tests, $n = 8$ for (**i**) and $n = 12$ for (**j**)). **k, l** Intraspinal injection of AAV-VEGFB-EGFP significantly reduced the withdrawal threshold and withdrawal latency 21 days after injection (*$P < 0.05$, **$P < 0.01$ vs. the corresponding AAV-EGFP group, two-tailed permutation tests, $n = 12$). The data are presented as the mean ± s.e.m. Source data is available as a Source Data file

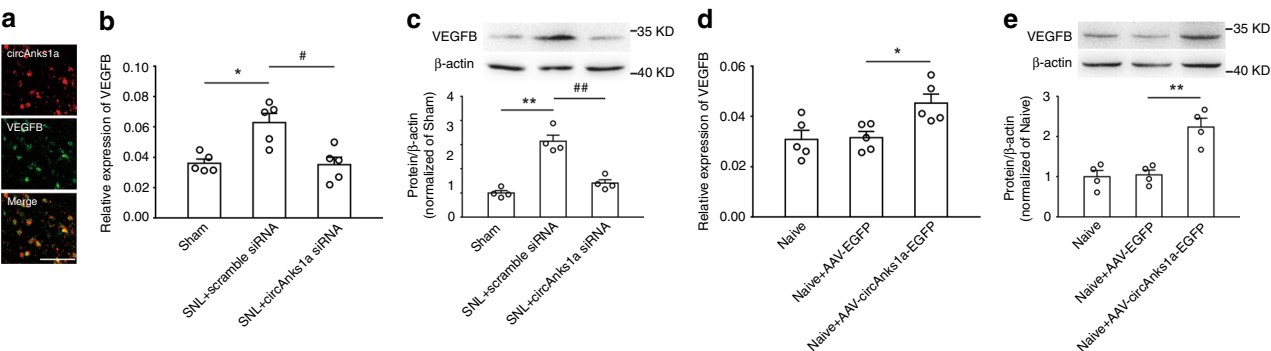

**Fig. 5** Upregulation of VEGFB was dependent on circAnks1a expression following SNL. **a** Colocalization of circAnks1a and VEGFB. Scale bar, 50 μm ($n = 3$). **b, c** Intrathecal injection of circAnks1a siRNA decreased the mRNA and protein expression of VEGFB in the spinal dorsal horn of rats on day 14 following SNL (*$P < 0.05$, **$P < 0.01$ vs. sham group, #$P < 0.05$, ##$P < 0.01$ vs. the corresponding scramble group, two-tailed one-way ANOVA, $n = 5$ for (**b**) and $n = 4$ for (**c**)). **d, e** Twenty-one days following intraspinal injection of recombinant AAV-circAnks1a-EGFP, the expression of VEGFB was significantly upregulated at mRNA and protein levels (*$P < 0.05$, **$P < 0.01$ vs. the AAV-EGFP group, two-tailed one-way ANOVA, $n = 5$ for (**d**) and $n = 4$ for (**e**)). The data are presented as the mean ± s.e.m. Source data are available as a Source Data file

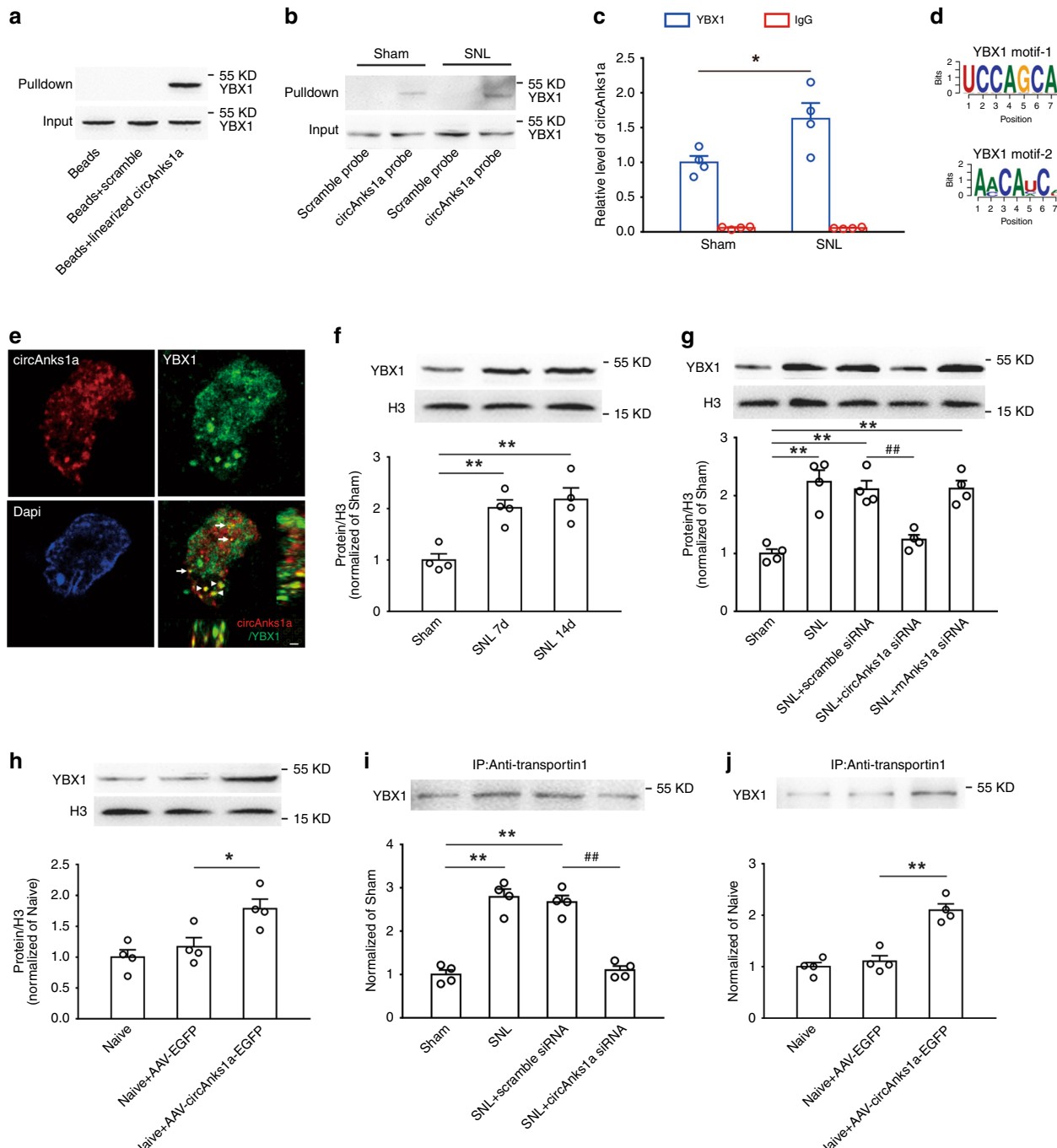

**Fig. 6** CircAnks1a promoted the nuclear translocation of YBX1 following SNL. **a** Exogenous linearized circAnks1a bound to YBX1 on day 14 following SNL ($n = 3$). **b** RNA pulldown assay showed that SNL significantly increased the interaction of endogenous circAnks1a and YBX1 in the dorsal horn ($n = 3$). **c** The amount of circAnks1a precipitated by YBX1 was obviously increased on day 14 following SNL (*$P < 0.05$ vs. the sham group, two-tailed two-sample $t$ tests, $n = 4$). **d** Two potential YBX1 motifs were observed in circAnks1a. **e** High-resolution image showing that binding of circAnks1a to YBX1 occurred in both the nucleus (arrow) and the cytoplasm (arrowhead). Scale bar, 1 μm ($n = 3$). **f** The level of nuclear YBX1 was significantly increased on days 7 and 14 after SNL (**$P < 0.01$ vs. the sham group, two-tailed one-way ANOVA, $n = 4$). **g** Suppression of circAnks1a by intrathecal injection of circAnks1a siRNA inhibited the increase in nuclear YBX1 on day 14 after SNL (**$P < 0.01$ vs. sham group, ##$P < 0.01$ vs. corresponding scramble group, two-tailed one-way ANOVA, $n = 4$). **h** Overexpression of circAnks1a by intraspinal injection of AAV-circAnks1a-EGFP increased the level of YBX1 in the nucleus (*$P < 0.05$ vs. AAV-EGFP group, two-tailed one-way ANOVA, $n = 4$). **i** CircAnks1a siRNA significantly attenuated the increase in YBX1 immunoprecipitation by transportin-1 antibody on day 14 after SNL (**$P < 0.01$ vs. sham group, ##$P < 0.01$ vs. corresponding scramble group, two-tailed one-way ANOVA, $n = 4$). **j** Intraspinal injection of AAV-circAnks1a-EGFP increased the amount of YBX1 in the immunocomplex precipitated by transportin-1 (**$P < 0.01$ vs. the AAV-EGFP group, two-tailed one-way ANOVA, $n = 4$). The data are presented as the mean ± s.e.m. Source data are available as a Source Data file

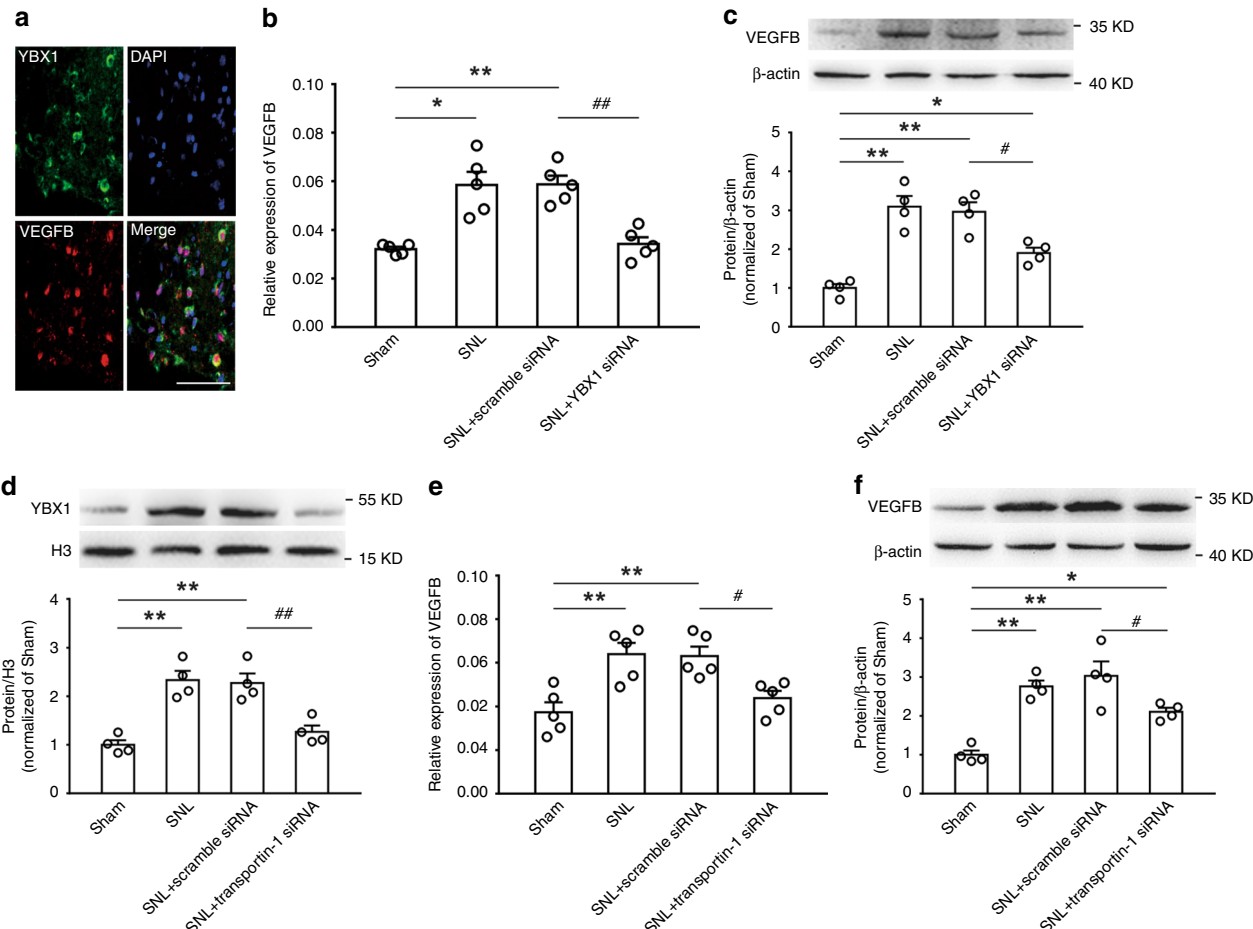

**Fig. 7** Increased YBX1 in the nucleus contributes to VEGFB expression. **a** Representative image labeled for YBX1, VEGFB, and DAPI. Scale bar, 50 μm (n = 3). **b** Continuous intrathecal administration of YBX1 siRNA prevented the upregulation of VEGFB mRNA induced by SNL ($^*P < 0.05$, $^{**}P < 0.01$ vs. sham group, $^{##}P < 0.01$ vs. corresponding scramble group, two-tailed one-way ANOVA, n = 5). **c** YBX1 siRNA (i.t.) attenuated the VEGFB protein upregulation on day 14 after SNL ($^*P < 0.05$, $^{**}P < 0.01$ vs. sham group, $^{#}P < 0.05$ vs. corresponding scramble group, two-tailed one-way ANOVA, n = 4). **d** Suppression of transportin-1 by transportin-1 siRNA (i.t.) decreased YBX1 accumulation in the nucleus ($^{**}P < 0.01$ vs. sham group, $^{##}P < 0.01$ vs. corresponding scramble group, two-tailed one-way ANOVA, n = 4). **e, f** Intrathecal injection of transportin-1 attenuated VEGFB upregulation on day 14 after SNL ($^*P < 0.05$, $^{**}P < 0.01$ vs. sham group, $^{#}P < 0.05$ vs. corresponding scramble group, two-tailed one-way ANOVA, n = 5 for (**e**) and n = 4 for (**f**)). The data are presented as the mean ± s.e.m. Source data are available as a Source Data file

group (Fig. 6i), and circAnks1a overexpression increased the YBX1 content of immunocomplexes precipitated by transportin-1 in the naïve group (Fig. 6j). Next, we verified the interaction between circAnks1and transportin-1. Immunoprecipitation with transportin-1 antibody using RIP methods showed that SNL significantly increased the level of circAnks1a, and this was suppressed by YBX1 siRNA (Supplementary Fig. 6e). Further EMSA showed that recombinant transportin-1 protein did not bind to circAnks1a site-1 (Supplementary Fig. 6f). Taken together, these results suggest that circAnks1a enhances the interaction between YBX1 and transportin-1 by interacting directly with YBX1, thus promoting nuclear translocation of YBX1 in the dorsal horn after nerve injury.

**YBX1 contributed to VEGFB upregulation.** It is well-known that the transcription factor YBX1 modifies the expression of target genes by interacting with DNA in the nucleus. We examined whether the increased nuclear YBX1 participated in the circAnks1a-mediated VEGFB upregulation that was observed following SNL. The results showed that YBX1 immunofluorescence was present in the DAPI-positive nuclear region and that the YBX1 immunofluorescence colocalized with VEGFB-positive cells

(Fig. 7a). Furthermore, intrathecal application of YBX1 siRNA completely blocked the upregulation of VEGFB mRNA (Fig. 7b), and partially attenuated the increase in VEGFB protein in SNL rats (Fig. 7c). Notably, administration of transportin-1 siRNA (i.t.) also attenuated the upregulation of YBX1 in the nucleus (Fig. 7d) and the increases in VEGFB mRNA (Fig. 7e) and protein (Fig. 7f) on day 14 following SNL.

**CircAnks1a promoted recruitment of YBX1 to *Vegfb* promoter.** To explore the mechanism underlying nuclear YBX1 mediation of VEGFB upregulation, we first predicted the potential binding sites of YBX1 in the *Vegfb* gene promoter region using the Genomatix database (Fig. 8a). ChIP-PCR assays using primers designed to amplify a fragment (−2036 to −1820) of the *Vegfb* promoter showed that recruitment of YBX1 to the *Vegfb* promoter was significantly increased on day 14 following SNL (Fig. 8b). To further confirm that the binding of YBX1 to the *Vegfb* promoter is functional, a 2200-bp fragment of the *Vegfb* promoter was fused to the promoter-less firefly luciferase gene of the pGL3-Basic vector to generate a *Vegfb*-luc reporter. Promoter activity was assessed by measuring luciferase activities in transfected 293T cells. YBX1 overexpression enhanced the luciferase

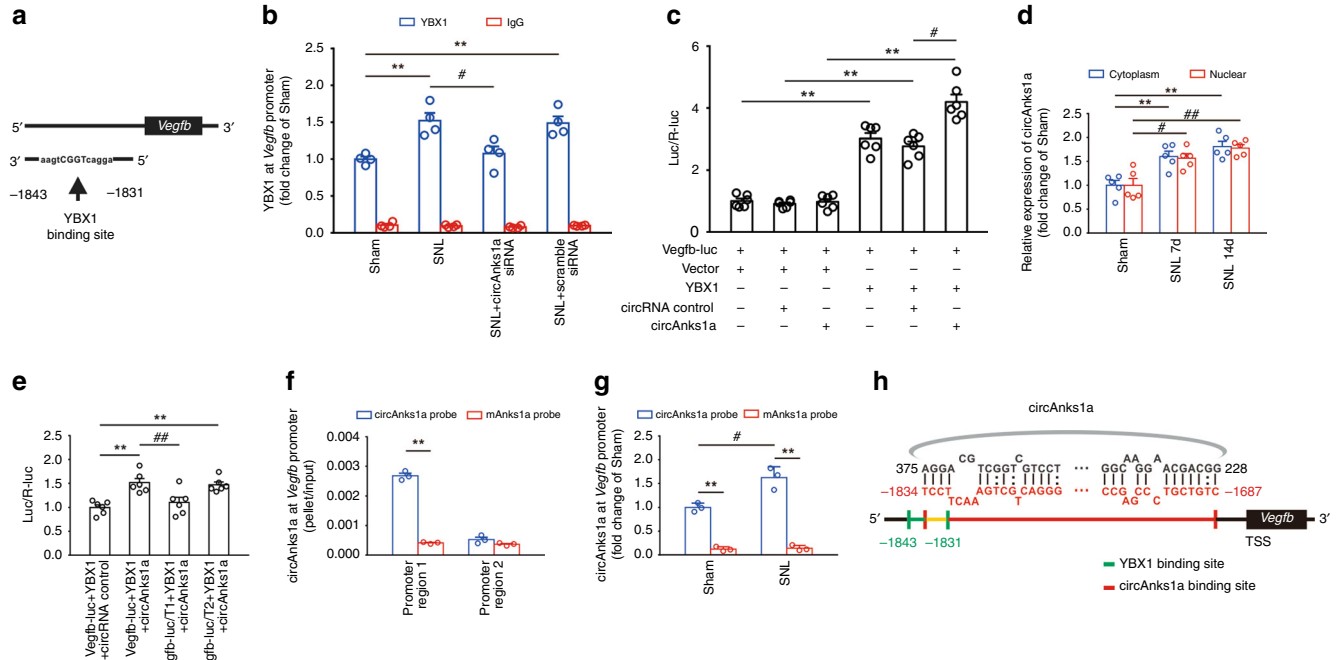

**Fig. 8** CircAnks1a binding to *Vegfb* facilitated the recruitment of YBX1 to the *Vegfb* promoter. **a** The simulation showed a potential YBX1 binding site in the *Vegfb* gene promoter region at position −1843/−1831. **b** Chromatin immunoprecipitation assays were performed using YBX1 antibody on day 14 after SNL with or without circAnks1a siRNA application (i.t.) (**P < 0.01 vs. sham group, #P < 0.05 vs. SNL + scramble siRNA group, two-tailed one-way ANOVA, n = 4). **c** YBX1 overexpression enhanced luciferase activities, while circAnks1a overexpression had no effect on luciferase activity. (**P < 0.01 vs. control vector, #P < 0.05 vs. circRNA control, two-tailed one-way ANOVA, n = 6). **d** CircAnks1a expression in the cytoplasm and nucleus was examined on days 7 and 14 after SNL (**P < 0.01 vs. cytoplasm in sham group, #P < 0.05, ##P < 0.01 vs. nucleus in sham group, two-tailed one-way ANOVA, n = 5). **e** Dual-luciferase reporter assays showed that circAnks1a promoted the transcription of *Vegfb* by YBX1 via position −1834 to −1687 (promoter region 1). T1 truncation of promoter region 1, T2 truncation of promoter region 2 (**P < 0.01 vs. circRNA control group, ##P < 0.01 vs. the corresponding *Vegfb*-luc group, two-tailed one-way ANOVA, n = 6). **f** CircAnks1a markedly bound to the *Vegfb* promoter at position −1834 to −1687 (promoter region 1) in circAnks1a-overexpressing C6 cells (**P < 0.01 vs. mAnks1a probe group, two-tailed two-sample t tests, n = 3). **g** Binding of circAnks1a to the *Vegfb* promoter at position −1834 to −1687 was significantly increased following SNL (**P < 0.01 vs. corresponding mAnks1a probe group, #P < 0.05 vs. sham group, two-tailed two-sample t tests, n = 3). **h** Simulated diagram showing the binding sites between circAnks1a and *Vegfb* (red) and the binding sites between YBX1 and *Vegfb* (green). The data are presented as the mean ± s.e.m. Source data are available as a Source Data file

activities compared with the negative control vector, while overexpression of circAnks1a alone had no effect on *Vegfb* transcription (Fig. 8c).

In addition, the PCR results showed that circAnks1a expression in the nucleus was significantly increased on days 7 and 14 following SNL (Fig. 8d). Considering the high-resolution images showing that circAnks1a binds to YBX1 in the neuronal nucleus (Fig. 6e), we tested the hypothesis that circAnks1a also affects the transcriptional regulation of *Vegfb* by YBX1. Our study showed that suppression of circAnks1a expression by intrathecal injection of circAnks1a siRNA significantly decreased the enrichment of YBX1 (Fig. 8b) and that overexpression of circAnks1a increased the enrichment of YBX1 on the *Vegfb* promoter (Supplementary Fig. 7). Importantly, the luciferase assay analysis showed that the binding of YBX1 to the *Vegfb* promoter was enhanced by overexpression of circAnks1a, indicating that circAnks1a may directly promote the recruitment of YBX1 to the *Vegfb* promoter (Fig. 8c, e). Moreover, in silico analysis using IntaRNA, LncRNA and LncTar showed that the *Vegfb* promoter has two potential circAnks1a binding regions, one at positions −1834 to −1687 (promoter region 1) and another at −1510 to −1451 (promoter region 2). We constructed promoter region 1- and promoter region 2-truncated dual luciferase vectors and found that the luciferase activities decreased significantly when promoter region 1 was truncated (Fig. 8e). Next, we performed chromatin isolation by RNA purification (ChIRP) assays in vitro and found that circAnks1a bound strongly to the *Vegfb* promoter at positions

−1834 to −1687, but that it did not bind strongly at positions −1510 to −1451 in circAnks1a-overexpressing C6 cells (Fig. 8f). These results suggest that circAnks1a promotes the transcription of *Vegfb* by YBX1 only when it is bound to promoter region 1. In in vivo experiments, we confirmed that SNL treatment significantly enhanced the binding of circAnks1a to the *Vegfb* promoter at positions −1834 to −1687 compared with the sham group (Fig. 8g). The above evidence demonstrates for the first time that exonic circAnks1a in the nucleus binds directly to the *Vegfb* gene and promotes the transcription of *Vegfb* by recruiting YBX1 to the *Vegfb* promoter (Fig. 8h).

**CircAnks1a acted as a sponge for miR-324-3p.** The above results showed that administration of YBX1 siRNA completely blocked the upregulation of VEGFB mRNA (Fig. 7b) but only partially attenuated the increase in VEGFB protein in SNL rats (Fig. 7c). Considering our finding that circAnks1a siRNA completely inhibited the upregulation of VEGFB protein induced by nerve injury, another YBX1-independent pathway might exist through which circAnks1a regulates the expression of VEGFB in the setting of nerve injury. It is well-known that miRNAs play a critical role in post-transcriptional regulation and that cytoplasmic circRNAs can act as miRNA sponges to regulate target gene translation. We first conducted RIP from spinal dorsal horn tissues using an antibody against argonaute 2 (AGO2). The qPCR analysis showed that endogenous circAnks1a was enriched in

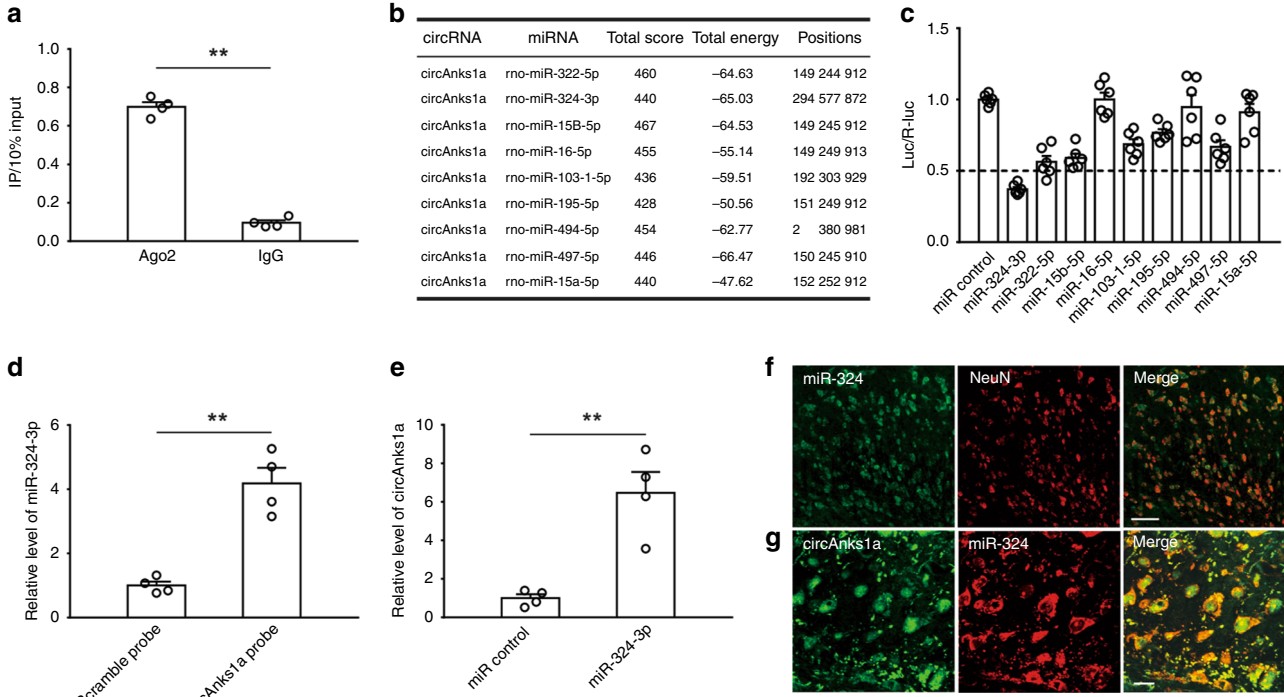

**Fig. 9** CircAnks1a interacted with miR-324-3p. **a** RIP showed that circAnks1a was enriched in AGO2 immunoprecipitates ([**]$P < 0.01$ vs. IgG group, two-tailed two-sample $t$ tests, $n = 4$). **b** 9 miRNAs had at least 3 potential sites in circAnks1a by in silico analysis. **c** A luciferase reporter assay for the luciferase activity of Luc-circAnks1a in 293T cells transfected with different miRNA mimics was used to identify miRNAs that bound to the circAnks1a sequence ($n = 6$). **d** Compared with the scramble probe, the level of miR-324-3p pulled-down by the circAnks1a probe was obviously higher ([**]$P < 0.01$ vs. scramble group, two-tailed two-sample $t$ tests, $n = 4$). **e** MiR-324-3p probe was used to examine the interaction between miR-324-3p and circAnks1a in dorsal horn tissues ([**]$P < 0.01$ vs. miR control group, two-tailed two-sample $t$ tests, $n = 4$). **f, g** FISH showed that miR-324-3p was expressed in neurons and that it colocalized with circAnks1a-positive cells. Scale bars, 50 and 10 μm, respectively ($n = 3$). The data are presented as the mean ± s.e.m. Source data is available as a Source Data file

AGO2 immunoprecipitates, implying that circAnks1a might have the ability to sponge miRNAs (Fig. 9a). Next, we used miRanda to predict that 84 miRNAs had potential binding sites with circAnks1a (Supplementary Table 4); 9 of these miRNAs had at least 3 potential binding sites for circAnks1a (Fig. 9b). After constructing a luciferase reporter by inserting the entire circAnks1a sequence into the 3' untranslated region (UTR) of firefly luciferase, we performed a luciferase screening by co-transfecting the miRNA mimic and the luciferase reporter into 293T cells. Compared with control miRNA, miR-324-3p significantly reduced the luciferase reporter activities by more than 50% (Fig. 9c). Next, we performed RNA pull-down experiments using dorsal horn tissue. The results showed that the circAnks1a-specific probe significantly enriched the miR-324-3p compared to the scramble probe (Fig. 9d). Furthermore, we used a biotin-coupled miR-324-3p mimic to test whether miR-324-3p could pull-down circAnks1a. The results showed that the biotin-coupled miR-324-3p probe captured more circAnks1a than the biotin-coupled control probe in pull-down experiments using dorsal horn tissues (Fig. 9e). FISH demonstrated that miR-324-3p was expressed in neurons (Fig. 9f) and that it colocalized with circAnks1a (Fig. 9g). Taken together, these results suggest that circAnks1a acts as a sponge for miR-324-3p.

**CircAnks1a regulates VEGFB expression through miR-324-3p.**
We observed that miR-324-3p colocalized with VEGFB in the dorsal horn (Fig. 10a), and we predicted a putative miR-324-3p binding sequence in the 3'UTR of VEGFB mRNA using miRanda and RNA22 (Fig. 10b). The luciferase reporter vector in which the

3'UTR of VEGFB mRNA was fused to the 3'UTR of luciferase was transfected into 293T cells together with miR-324-3p mimics. The results showed that miR-324-3p mimics significantly reduced the luciferase signal compared with the control miRNA (Fig. 10c). In vivo experiments further showed that intrathecal injection of miR-324-3p agomir significantly attenuated the increase in VEGFB protein (Fig. 10d), but did not affect VEGFB mRNA levels (Fig. 10e) on day 14 following SNL. Behavioral tests also showed that miR-324-3p agomir application (i.t.) significantly increased the withdrawal threshold (Fig. 10f) and the withdrawal latency (Fig. 10g) in SNL-treated rats. Furthermore, suppression of miR-324-3p by intrathecal injection of antagomir did not change VEGFB mRNA levels (Fig. 10h) but markedly promoted the expression of the protein in naïve animals (Fig. 10i). Importantly, miR-324-3p antagomir treatment significantly induced mechanical allodynia (Fig. 10j) and thermal hyperalgesia (Fig. 10k) in naïve animals. To verify that circAnks1a regulates VEGFB expression by targeting miR-324-3p, we co-transfected the circAnks1a vector and miR-324-3p mimics into 293T cells containing the luc-VEGFB vector. The results showed that circAnks1a overexpression rescued the decrease in the luciferase signal in the presence of miR-324-3p mimics (Fig. 10l). These results support the conclusion that circAnks1a also regulates the expression of VEGFB translationally by sponging miR-324-3p following SNL.

**Discussion**
In the present study, we identified for the first time a spinal cord-specific conservative circRNA, circAnks1a, that may be a key

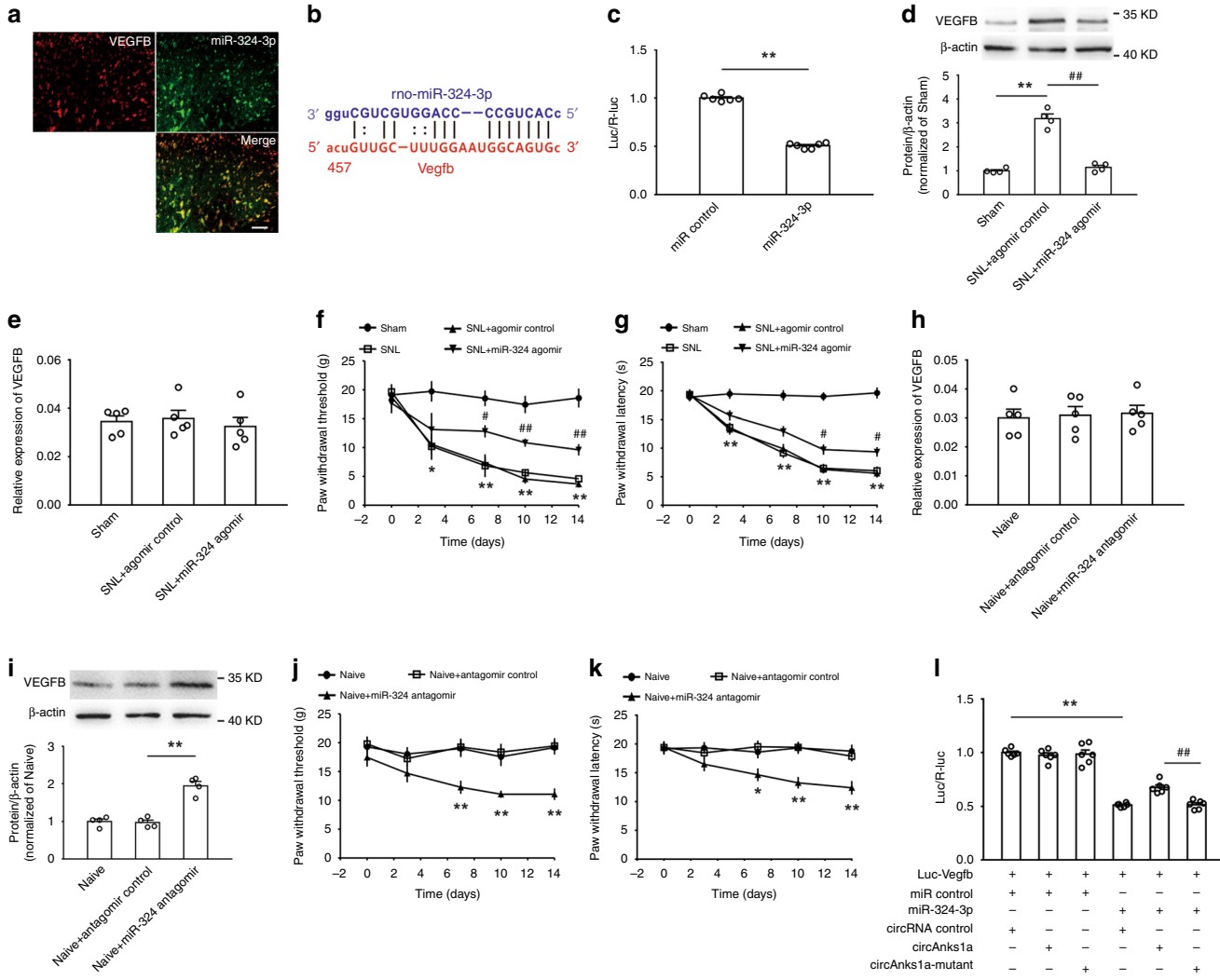

**Fig. 10** CircAnks1a regulates VEGFB expression by targeting miR-324-3p following SNL. **a** MiR-324-3p colocalized with VEGFB-positive cells. Scale bar, 50 μm ($n = 3$). **b** A putative binding site between miR-324-3p and the 3'UTR of VEGFB mRNA was predicted. **c** Administration of miR-324-3p mimics significantly suppressed luciferase activity (**$P < 0.01$ vs. miR control group, two-tailed two-sample $t$ tests, $n = 6$). **d** The effect of miR-324-3p agomir (i.t.) on VEGFB protein levels on day 14 after SNL (**$P < 0.01$ vs. sham group, ##$P < 0.01$ vs. corresponding agomir group, two-tailed one-way ANOVA, $n = 4$). **e** Intrathecal injection of miR-324-3p agomir did not affect VEGFB mRNA expression on day 14 after SNL ($n = 5$). **f, g** Intrathecal injection of miR-324-3p agomir attenuated mechanical allodynia and thermal hyperalgesia (*$P < 0.05$, **$P < 0.01$ vs. sham group, #$P < 0.05$, ##$P < 0.01$ vs. corresponding agomir group, two-tailed permutation tests, $n = 12$). **h** The effect of miR-324-3p antagomir on VEGFB mRNA ($n = 5$). **i** Intrathecal injection of miR-324-3p antagomir increased VEGFB protein expression (**$P < 0.01$ vs. the corresponding antagomir group, two-tailed one-way ANOVA, $n = 4$). **j, k** The effect of miR-324-3p antagomir on mechanical allodynia and thermal hyperalgesia (**$P < 0.01$ vs. the corresponding antagomir group, two-tailed permutation tests, $n = 12$). **l** MiR-324-3p mimics significantly decreased the luciferase activity, and the circAnks1a overexpression vector rescued the reduced luciferase signal (**$P < 0.01$ vs. miR control group, ##$P < 0.01$ vs. circAnks1a mutant group, two-tailed one-way ANOVA, $n = 6$). The data are presented as the mean ± s.e.m. Source data are available as a Source Data file

biomarker in the treatment of chronic pain. Suppression of circAnks1a decreased VEGFB upregulation and attenuated the excitability of dorsal horn neurons and neuropathic pain following SNL, while overexpression of circAnks1a reduced the withdrawal threshold and withdrawal latency and increased VEGFB expression in naïve rodents. These data suggest for the first time that the novel circAnks1a/VEGFB pathway plays an important role in the development of neuropathic pain induced by nerve injury. Next, we found that the transcription factor YBX1 is a critical protein mediator of circAnks1a-regulated VEGFB expression in rodents with neuropathic pain. The increased binding of circAnks1a to YBX1 in the cytoplasm promoted the translocation of YBX1 into the nucleus through

transportin-1. We further found that nuclear circAnks1a can bind to the promoter region of *Vegfb* following SNL treatment and that this enhanced the recruitment of transcription factor YBX1 to the *Vegfb* promoter and facilitated *Vegfb* transcription. Taken together, our findings indicated that circAnks1a facilitated the nuclear translocation and promoter binding of YBX1, thereby inducing VEGFB upregulation and neuropathic pain induced by nerve injury.

According to their biogenesis, circRNAs can be classified into three categories: exonic circular RNAs, intronic circular RNAs and exon–intron circular RNAs[5,27,28]. Previous studies showed that exonic circular RNAs are mainly distributed in the cytoplasm. Here, we found that circAnks1a, as an exonic circular

RNA, was expressed in both the cytoplasm and the nucleus of dorsal horn neurons. Furthermore, circAnks1a was specifically expressed in the spinal dorsal horn only, and it was significantly increased in rodents with nerve injury. These data raise the possibility that circAnks1a might exhibit remarkable regulatory function and mediate the development of chronic pain induced by nerve injury. This possibility was further validated by our study of the gain and loss of circAnks1a function in modeled and naive rodents. We found that silencing endogenous circAnks1a with siRNA significantly attenuated the neuronal hyper-excitablity, mechanical allodynia and thermal hyperalgesia that otherwise occur in rats with nerve injury and that overexpression of circAnks1a enhanced glutamatergic transmission in dorsal horn neurons and induced hypersensitivity to mechanical and thermal stimuli in control rodents. Studies have reported changes in the expression of some circRNAs (>twofold) in rats with sciatic chronic constriction injury (CCI)[15,16], and changes in the expression of several circRNAs, such as circRnf4, were shown to overlap between CCI and SNL, although some differences existed. The difference in circAnks1a expression in these studies and in the current study may result from different experimental conditions or detection methods. The present study for the first time identified the functional involvement of a spinal cord-specific and conservative circular RNA, namely, circAnks1a, in the regulation of neuronal excitability and pain behavior in rodents with nerve injury; thus, circAnks1a provides an attractive biomarker for treating chronic pain.

We further identified VEGFB as the target protein whose expression in the dorsal horn was meticulously regulated, at the transcriptional and translational levels, by circAnks1a, thus contributing to the central sensitization and the pain behavior induced by nerve injury. While VEGFB released from tumors is known to augment pain sensitivity through selective activation of VEGFR1 expression in peripheral sensory neurons in cancer pain models[29], the present study for the first time illustrated the functional significance of spinal neuron-derived VEGFB in the central sensitization to neuropathic pain induced by spinal nerve injury. We first observed increased expression of VEGFB in the dorsal horn with a time course consistent with behavior hypersensitivity induced by nerve injury. Local knockdown or deletion of VEGFB using VEGFB siRNA and local deletion of VEGFB by injection of AAV-Cre into VEGFB[flox/flox] mice decreased the excitability of dorsal horn neurons and ameliorated the neuropathic pain induced by nerve injury, whereas over-expression of VEGFB induced hypersensitivity to mechanical and thermal stimuli in normal rats. The binding of VEGFB to its receptor leads to activation of a number of downstream pathways, including p38 MAPK, ERK/MAPK, PKB/AKT, and PI3K[19,30]. It is well-known that activation of p38 MAPK or PKB/AKT can enhance neuronal excitability via the initiation of an inflammatory response in the nervous system. Importantly, the upregulation of VEGFB coincided with the increase in circAnks1a in dorsal horn neurons, and suppression of circAnks1a by specific siRNA inhibited VEGFB upregulation in dorsal horn neurons as well as pain behavior induced by SNL. Together, these results demonstrated that increased levels of VEGFB in dorsal horn neurons, which are subject to regulation by circAnks1a, contribute substantially to the central sensitization and pain behavior induced by nerve injury.

Next, we determined the mechanism through which circAnks1a mediates the transcriptional regulation of VEGFB induced by nerve injury. We found that spinal nerve injury significantly increased the interaction between circAnks1a and transcription factor YBX1, which regulates the expression of target genes via translocation into the nucleus and binding to promoter regions[31]. A previous study reported that lncRNA-

BX111 could recruit YBX1 to the ZEB1 promoter, consequently activating ZEB1 transcription[32]. Similarly, the present study showed that circAnks1a facilitated the nuclear translocation of YBX1 and its binding to the *Vegfb* promoter region, thereby regulating VEGFB expression and pain behavior induced by nerve injury. SNL significantly increased the interaction between YBX1 and transportin-1, an essential step in YBX1 transloca-tion[26], and the nuclear distribution of YBX1 in dorsal horn neurons, and this was attenuated by intrathecal injection of circAnks1a siRNA. Furthermore, luciferase assays showed that the binding of YBX1 to the *Vegfb* promoter was enhanced by overexpression of circAnks1a. Importantly, chromatin isolation by RNA purification (ChIRP) using circAnks1a probes showed that circAnks1a exhibited marked binding to the *Vegfb* promoter in vitro, and SNL enhanced the binding compared with that observed in sham rats. In addition, inhibition of circAnks1a attenuated the increased binding of circAnks1a and YBX1 to the *Vegfb* promoter observed in rats with nerve injury. Taken together, the present study for the first time demonstrated that enhanced interaction between circAnks1a and YBX1 can facilitate YBX1 nuclear translocation and its binding to the *Vegfb* promoter region and thereby lead to VEGFB upregulation induced by nerve injury. Whether or not circAnks1a regulates the expression of other genes that participate in chronic pain needs further study.

We further characterized the circAnks1a-mediated miRNA regulation of VEGFB mRNA expression and its functional involvement in pain behavior induced by nerve injury. Recent studies have indicated that circRNAs function as miRNA spon-ges to modulate the pathogenesis and progression of various diseases[33,34]. In the present study, we first showed, using a luciferase reporter assay, that circAnks1a binds to miR-324-3p. RNA pull-down and FISH analysis further showed that cir-cAnks1a interacts with miR-324-3p in the dorsal horn to regulate neuropathic pain induced by SNL. It is well-known that dys-gulation of miRNA expression in the spinal cord regulates the expression of target proteins and contributes to the development of neuropathic pain[35]. Previous study showed that miR-324, by downregulating potassium channel Kv4.2 expression, increased the excitability of hippocampal neurons and cardiac cells[36]. In the present study, we found that miR-324-3p targeted VEGFB mRNA and regulated VEGFB expression and pain behavior in rodents with nerve injury. Bioinformatics analysis also showed that miR-324-3p can target VEGFB mRNA in humans, sug-gesting the potential involvement of miR-324-3p-mediated VEGFB expression in patients with neuropathic pain. We fur-ther found that circAnks1a binds to miR-324-3p and that these two RNAs are colocalized in dorsal horn neurons. Considering that circAnks1a functions to mediate VEGFB upregulation and pain behavior induced by nerve injury, miR-324-3p sponging activity by circAnks1a potentially contributed to VEGFB upre-gulation and subsequent central sensitization and pain behavior induced by nerve injury.

Taken together, our results illustrate the molecular mechanism through which circAnks1a regulates VEGFB expression in dorsal horn neurons at transcriptional and posttranscriptional levels and contributes to central sensitization and pain behavior induced by nerve injury. Interaction between circAnks1a and YBX1 promoted YBX1 translocation into the nucleus and enhanced its binding to the *Vegfb* promoter, thus facilitating *Vegfb* transcription in dorsal horn neurons following nerve injury. The increase in the levels of circAnks1a also sponged miR-324-3p, thereby unleashing VEGFB mRNA translation in the dorsal horn. These findings identify novel targets for the development of effective treatment for neuropathic pain induced by nerve injury.

## Methods

**Animals.** Male Sprague–Dawley rats weighing 200–220 g were obtained from the Institute of Experimental Animals of Sun Yat-sen University. The mice used in this study were adult males of C57BL/6 background weighing 20–30 g. VEGFB$^{flox/flox}$ mice were graciously provided by Prof. Xu-Ri Li of the Zhongshan Ophthalmic Center. All animals were kept at 24 °C and 50–60% humidity under a 12:12-h light/dark cycle and with ad libitum access to food and water. All experimental procedures were approved by the Local Animal Care Committee and were conducted in accordance with the guidelines of the National Institutes of Health (NIH) on animal care and with the ethical guidelines.

**CircRNA sequencing.** Total RNA samples (2 μg) were treated with the Epicenter Ribo-Zero rRNA Removal Kit (Illumina) and RNase R (Epicenter) to remove ribosomal and linear RNA. Then, RNA-seq libraries were constructed using the NEBNext® Ultra™ RNA Library Prep Kit for Illumina® (NEB) according to the manufacturer's instructions. The libraries were quality controlled with Agilent 2200 TapeStation (Agilent) and sequenced using Illumina HiSeq 3000 by RiboBio Biotechnology Co (Guangzhou, China).

**Bioinformatics analysis.** For identification and quantification of circRNAs, reads obtained from the sequencing were first filtered to obtain high-quality clean reads. To obtain effective clean reads, the residual rRNA was mapped and removed using an RNA central database. The remaining reads were used for alignment and analysis. For each sample, the rRNA-removed reads were first mapped to the rat reference genome (Rnor_6.0) by TopHat2. After alignment with the reference genome, the unmapped reads were then remapped using the Tophat-fusion module to identify circRNAs[37]. The identified circRNA was called if it was supported by at least one unique back-spliced read in at least one sample. Next, we used HTseq to calculate the count values and RPM values of the circRNAs in each sample and subjected the candidate circRNAs to annotation and length analysis. Differential expression analysis of the RNA-seq was performed using DESeq in R with the default parameters.

The potential binding sites of circAnks1a to the *Vegfb* promoter were predicted by three bioinformatics software programs (IntaRNA, LncRNA and LncTar). LncRNA (LongTarget) was developed to predict ncRNA-DNA binding motifs and binding sites in a genomic region based on potential base pairing rules; the LncRNA (LongTarget) analysis showed that a specific region of the *Vegfb* promoter (−1451 to −1510) was a potential binding site for circAnks1a. The results of IntaRNA analysis, a program for the accurate prediction of interactions between two RNA molecules, indicated that the region of circAnks1a from 228 to 375 may be a site with high affinity for the −1834 to −1687 region of the *Vegfb* promoter. Finally, LncTar calculation showed that circAnks1a could bind to the region of −1041 to −2000 of the *Vegfb* promoter. The binding of miRNA to circAnks1a and the *Vegfb* gene was predicted by miRanda.

**Whole-genome expression microarray analysis.** Total RNAs were reverse transcribed into double-stranded cDNAs. The cDNAs were invitro transcribed into antisense cRNAs and labeled with Cy3-CTP and Cy5-CTP using a Two-Color Low Input Quick Amp Labeling Kit (Agilent Technologies). Fluorescent dye-labeled cRNAs were fragmented and hybridized to a Sure Print G3 Rat GE 8 × 60 K microarray using an Agilent Gene Expression Hybridization Kit. The fluorescence intensities at 635 nm (Cy5) and 532 nm (Cy3) were measured using an Agilent microarray scanner. The microarray data were extracted using Agilent Feature Extraction Software. Low-intensity spots were removed so that each gene had expression values in more than 80% of the samples analyzed. Signals were normalized by Loess normalization. Differentially expressed genes (DEGs) were screened using SAM v4.01 software with the false discovery rate set to 5% ($q$-value < 0.05).

**RNA preparation and qRT-PCR.** Nuclear and cytoplasmic RNA was extracted using Norgen's Cytoplasmic & Nuclear RNA Purification Kit (Norgen Biotek Corp.). RNAs were extracted using Trizol reagent (Life Technologies). For RNase R treatment, 2 μg of RNA was incubated with 3 U μg$^{−1}$ of RNase R (Epicenter) for 20 min at 37 °C. To quantify the amount of mature miRNA, the All-in-One™ miRNA qRT-PCR Detection Kit (GeneCopoeia) was used according to the manufacturer's instructions. PrimeScript RT Master Mix (Takara) and SYBR Premix Ex Taq II (Takara) were used to quantify the amounts of mRNA and circRNA. In particular, the annealing of divergent primers at the distal ends of circRNAs was used to determine the abundance of circRNA. The primers used are listed in Supplementary Table 5.

**SNL and behavioral tests.** In rats under isoflurane (4%) anesthesia, the left L5 spinal nerve was isolated adjacent to the vertebral column and tightly ligated with 6–0 silk sutures distal to the dorsal root ganglion and proximal to the formation of the sciatic nerve. In sham-operated rats, the L5 spinal nerves were identically exposed without ligation.

For behavioral testing, animals were placed in a plastic box for 3 consecutive days (15 min day$^{−1}$) to allow them to adapt to the environment before testing. Von Frey filaments that produce different forces were applied alternately to the plantar

surface of the hind paw. In the absence of a paw withdrawal response, a stronger stimulus was presented; when paw withdrawal occurred, the next weaker stimulus was chosen. Optimal threshold calculation by this method required 5 responses in the immediate vicinity of the 50% threshold.

Thermal hyperalgesia was tested using a plantar test (Ugo Basile Plantar Test Apparatus). Briefly, a radiant heat source beneath a glass flour was aimed at the plantar surface of the hind paw. Three measurements of hind paw withdrawal latency were taken for each hind paw and averaged as the result of each test. A 25-s cutoff was set to prevent tissue damage. All the experiments were performed by investigators who were blinded to the treatments/conditions.

**Intrathecal or intraspinal injection and AAV construction.** The cholesterol-conjugated siRNA, miR-324-3p agomir and antagomir used for intrathecal injection were obtained from RiboBio. A polyethylene-10 catheter was implanted into the L5/L6 intervertebral subarachnoid space after the injection of sodium pentobarbital (50 mg kg$^{−1}$, i.p.); the localization of the tip of the catheter was between the levels of the L4–L6 spinal segments. The rats were allowed to recover for 5 days. Animals that exhibited hind limb paresis or paralysis were excluded from the study. SiRNA (specific sequences are listed in Supplementary Table 6) and miR-324-3p agomir (1 nmol day$^{−1}$) were intrathecally injected 30 min prior to SNL surgery, and administration was continued for 5 days. MiR-324-3p antagomir (1 nmol day$^{−1}$) was intrathecally injected for 5 consecutive days.

For intraspinal injection of the AAV virus, the L4–L5 vertebrae were exposed, and the vertebral column was mounted in a stereotaxic frame. A slight laminotomy was performed, and the dura was incised to expose the spinal cord. AAV virus was injected into both sides of the spinal dorsal horn at 4 injection sites (150 nl of AAV was injected at each site). The micropipette was withdrawn 10 min after viral injection, and the incision was closed with stitches.

AAV2/9 viruses were designed and constructed by standard methods with assistance from Obio Technology (Shanghai, China) or BrainVTA (Wuhan, China). Briefly, a modified vector, pAAV-hSyn-polyA, was obtained by replacing the DNA fragment between hSyn and the right ITR with PCR-amplified polyA and the flanking cloning sites. The target gene was synthesized and inserted following hSyn. A total of 293 cells were transfected with pAAV-RC and pHelper together with the pAAV vector. Forty-eight hours after transfection, virus was purified from the cell lysate using a heparin-agarose column and concentrated to its final volume.

**C-fiber-evoked action potentials in vivo.** The left sciatic nerve of rats was dissected free for bipolar electrical stimulation. C-fiber-evoked responses in the spinal dorsal horn (L4–L6 segments) were recorded at a depth of 50–500 μm from the dorsal surface with a microelectrode (impedance, 1–2 MΩ; exposed tip diameter, 1–2 μm). A bandwidth of 0.1–500 Hz was used to record field potentials. An A/D converter card (DT2821-F-16SE, Data Translation) was used to digitize and store data in a Pentium computer at a sampling rate of 10 kHz. Single square pulses (0.5 ms duration at 1-min intervals) delivered to the sciatic nerve were used as test stimuli. The numbers of action potentials were counted at different simulation intensities. The distance from the stimulation site at the sciatic nerve to the recording site in the lumbar spinal dorsal horn was approximately 11 cm.

**Spinal cord slice preparation.** The L4–L6 spinal cord was quickly removed from the lumbar vertebrae and transferred to oxygenated (95% O$_2$ and 5% CO$_2$) ice-cold slice solution containing (in mM): 126 NaCl, 3 KCl, 10 D-glucose, 26 NaHCO$_3$, 1.2 NaH$_2$PO$_4$, 0.5 CaCl$_2$, and 5 MgCl$_2$. The dorsal and ventral roots were carefully removed except that in some slices the associated dorsal roots were kept for reception of stimulation from the stimulus isolator. The spinal cord was coated with agarose (Sigma, USA), and 400-μm thick acute spinal L4–L6 cord slices were cut on a vibratome (Leica VT-1000 S). The slices were incubated in continuously oxygenated standard artificial cerebrospinal fluid (ACSF: 125 mM NaCl, 3 mM KCl, 26 mM NaHCO$_3$, 1.25 mM NaH$_2$PO$_4$, 2 mM CaCl$_2$, 1 mM MgCl$_2$ and 10 mM D-glucose, pH 7.3) for at least 1 h at 33 °C and then transferred to the recording chamber. Lamina I to II neurons were visualized using a 40× water-immersion objective on an upright infrared Nikon microscope (Nikon, Japan).

**Whole-cell recordings.** The recording chamber was continuously perfused with pre-heated 33 °C ACSF at a rate of 2 ml min$^{−1}$. The pipettes (3–6 MΩ, ~2 μm tip diameter) were pulled from borosilicate glass with filament (OD: 1.2 mm, ID: 0.69 mm) on a P-2000G micropipette puller (Sutter Instruments, USA). In the experiments involving recording of evoked responses, a concentric bipolar electrode (FHC, Bowdoin, ME USA) connected to a constant-current stimulus isolator (DS3; Digitimer Ltd., UK) was used to stimulate the spinal cord dorsal root. Data were recorded using an EPC 10 amplifier (HEKA Elektronik, Germany). Stimulus delivery and data acquisition were performed using Patchmaster software (HEKA Elektronik). A seal resistance of ≥ 2 GΩ and an access resistance of ≤20 MΩ were considered acceptable. The electrophysiological data were replaced and analyzed by Clampfit10.4 (Axon Instruments Inc., USA) and mini Analysis program6.0.7 (Synaptosoft Inc., Decatur, GA, USA).

**mEPSC recordings.** The neurons in lamina I/II of L4-6 spinal cord dorsal horn were voltage clamped at −70 mV, and mEPSCs were recorded after application of

TTX (0.5 μM) and picrotoxin (100 μM). The pipette contained an internal solution (135 mM K-gluconate, 0.5 mM CaCl₂, 2 mM MgCl₂, 5 mM EGTA, 5 mM HEPES, and 5 mM Mg-ATP, pH 7.3). To stain the recorded neurons, 0.5% biocytin (Sigma, USA) was also included in the internal solution. Current traces were recorded continuously for a period of 5 min after at least 5 min achieving whole-cell configuration and analyzed using mini Analysis program 6.0.7 (Synaptosoft Inc., Decatur, GA, USA).

**Action potential (AP) recordings**. The neurons in lamina I/II of the dorsal horn of L4–6 spinal cord were recorded using pipettes containing an internal solution (135 mM K-gluconate, 0.5 mM CaCl₂, 2 mM MgCl₂, 5 mM EGTA, 5 mM HEPES, 5 mM Mg-ATP, and 0.5% biocytin, pH 7.3) under current clamp conditions. All the recordings were clamped at −70 mV. APs were evoked by current injection every 10 s with step intervals of 20 pA from −80 to 400 pA over a period of 500 ms. The relationship between frequency and injected current was analyzed using Clampfit10.4 (Axon Instruments Inc., USA).

**EPSP-spike coupling (E–S) recordings**. Whole-cell recordings were acquired from neurons in lamina I/II of L4–6 spinal dorsal horn while a concentric bipolar electrode connected to a constant-current stimulus isolator was used to stimulate the spinal cord dorsal root under current clamp conditions with picrotoxin (100 μM) in ASCF. All recordings were performed for at least 5 min after achieving whole-cell configuration. The pipette contained an internal solution (135 mM K-gluconate, 0.5 mM CaCl₂, 2 mM MgCl₂, 5 mM EGTA, 5 mM HEPES, 5 mM Mg-ATP, and 0.5% biocytin, pH 7.3). The E–S protocol began with a 5-min stable baseline in which a 5-mV EPSP could be evoked by the stimulus isolator. The stimulus current at baseline was the base stimulus intensity, called N, for each cell. E–Ss were then recorded following a current stimulus at step intervals of 0.5 N from 1.0 to 2.5 N; the stimuli consisted of five bursts of five bursts of five bursts of five pulses each at 20 Hz. E–Ss were analyzed in Clampfit10.4 (Axon Instruments Inc., USA). Neurons transfected with AAV virus were visualized using a 40× water-immersion objective on an upright infrared Nikon microscope (Nikon, Japan). One neuron per slice was sampled for electrophysiological recording. The recorded neurons were NK1R+ projection neurons, as shown by the fact that the recorded biocytin-positive cells colocalized with NK1R-positive cells (Supplementary Fig. 2d).

**Liquid chromatography–mass spectrometry analysis**. The full-length sequence of circAnks1a was constructed into a linear DNA template containing the T7 promoter. A biotinylated RNA probe was transcribed in vitro using T7 RNA polymerase (Thermo Scientific) and bound to streptavidin C1 magnetic beads (Invitrogen). Spinal dorsal horn tissues were ground in liquid nitrogen and incubated in lysis buffer [50 mM Tris-HCl, 150 mM NaCl, 2 mM MgCl₂, 1% NP40, SUPERase-In (Ambion), and protease inhibitors (Roche)] on ice for 30 min. The lysates were then incubated with the RNA probe for 2 h at 4 °C. The beads were briefly washed five times with wash buffer and boiled in sodium dodecyl sulphate (SDS) buffer. The retrieved proteins were separated via SDS-PAGE. The gel bands were manually excised and digested with mass spectrometry-grade trypsin (Promega). The digested peptides were analyzed on an AB Sciex TripleTOF® 6600 System (AB Sciex). The mass spectrometry data were analyzed and identified using Mascot (Matrix Science) in the NCBI *Rattus* database and the UniProt *Rattus* database.

**RNA pull-down assays**. Biotinylated circAnks1a and miRNA probes were synthesized by Ribo Bio (Guangzhou, China). Spinal dorsal horn tissues were ground in liquid nitrogen and incubated in lysis buffer [50 mM Tris-HCl, pH 7.4, 150 mM NaCl, 2 mM MgCl₂, 1% NP40, SUPERase-In (Ambion), and protease inhibitors (Roche)] on ice for 30 min. The lysates were incubated with the biotinylated probes at RT for 4 h; streptavidin C1 magnetic beads (Invitrogen) were then added to each binding reaction, and the mixtures were incubated at RT for 2 h. The beads were then washed briefly with wash buffer [0.1% SDS, 1% Triton X-100, 2 mM EDTA, 20 mM Tris-HCl, and 500 mM NaCl] five times. The bound proteins in the pull-down were analyzed by western blotting. The bound RNA in the pull-down was further identified after purification and reverse transcription. The sequence of circAnks1a and miR-324-3p probes are shown in Supplementary Table 7a.

**RNA-binding protein immunoprecipitation**. RIP experiments were performed using a Magna RIP RNA-Binding Protein Immunoprecipitation Kit (Millipore). Briefly, the spinal dorsal horn was collected and homogenized into a single-cell suspension in ice-cold PBS. After centrifugation, the pellet was resuspended in an equal volume of RIP lysis buffer. Magnetic beads were incubated with 5 μg antibody against Ago2 (Abcam; ab32381; 5 μg), YBX1 (Abcam; ab12148; 5 μg) or IgG at RT. The tissue lysates were then incubated with the bead-antibody complexes overnight at 4 °C. After treatment with proteinase K, the immunoprecipitated RNAs were extracted and reverse-transcribed. The abundance of circAnks1a was detected by qPCR.

**Chromatin immunoprecipitation (ChIP)**. ChIP assays were performed using the ChIP Assay Kit (Thermo). The animal's L4 and L5 spinal cord was removed quickly and placed in 1% formaldehyde for 10 min at room temperature to cross-link transcription factors with chromatin. The formaldehyde was then inactivated by addition of 125 mM glycine. Sonicated chromatin extracts containing DNA fragments were immunoprecipitated using YBX1 antibody (Abcam; ab12148; 10 μg) or normal rabbit IgG and pre-blocked protein G-Sepharose beads overnight at 4 °C. The next day, the chromatin-protein-antibody-bead complexes were eluted, and the DNA was extracted. The precipitated DNA was resuspended in nuclease-free water, and qPCR was performed as described above. Finally, the ratio of ChIP/input in the spinal dorsal horn was calculated.

**Chromatin isolation by RNA purification (ChIRP)**. For affinity capture of complexes containing circAnks1a and chromatin, we designed probes covering the sequence of circAnks1a for use in ChIRP[38–40]. The animals' L4 and L5 spinal cord tissue was removed quickly and cut into pieces in cold PBS (0.1 M). The tissues or the collected C6 cells were crosslinked with 1% glutaraldehyde at 25 °C for 10 min and quenched by the addition of 1.25 M glycine for 5 min. Tissue pellets were collected by centrifugation and lysed in a buffer containing 50 mM Tris 7.0, 10 mM EDTA, 1% SDS, 1 mM PMSF, 1× protease inhibitor and 0.1 U μl⁻¹ RNase inhibitor. The lysates were sonicated to shear the DNA to lengths of 100–500 bp using an Ultrasonic Broken Instrument in a 4 °C water bath for 30 min. Meanwhile, pre-binding probes labeled with biotin (1 μl of 100 pmol μl⁻¹ probe per 1 ml chromatin) were incubated with streptavidin beads for 30 min. The sonicated samples were centrifuged, and the resulting supernatants were hybridized with the beads bound to probes in a buffer consisting of 750 mM NaCl, 1% SDS, 50 mM Tris 7.0, 1 mM EDTA, 15% formamide, 1 mM PMSF, 1× protease inhibitor and 0.1 U μl⁻¹ RNAse inhibitor at 37 °C overnight on an end-to-end shaker. Subsequently, the beads were washed five times with 1 ml of pre-warmed wash buffer for 5 min per wash at 37 °C. At the last wash step, 1/20 of the beads were reserved for qPCR analysis. DNA was extracted and used for qPCR analysis. The probe sequences are shown in Supplementary Table 7b.

**FISH and immunohistochemistry**. Animals were perfused through the ascending aorta with 4% paraformaldehyde under anesthesia. The spinal cord tissues were cut into 25-μm-thick transverse sections after 30% DEPC-sucrose dehydration at 4 °C and hybridized at 42 °C for 18 h with the 3' and 5'-TYE563-labeled circ-Anks1a probe 5'-TCGTTGTCGTTGTTCTTCAGT-3' (1:100, EXQON) and the 3' and 5'-TYE665-labeled miR-324-3p probe 5'-CAGCAGCACCTGGGGCA-3' (1:100, EXQON). The sections were then incubated at 4 °C overnight with primary antibodies against VEGFB (Santa Cruz; sc-80442; 1:50), YBX1 (Biorbyt; orb8480; 1:50), GFAP (CST; 3670; 1:200), Iba1 (Abcam; ab5076; 1:200), or NeuN (Millipore; MAB377; 1:200). After that, the sections were incubated with Cy3, Cy5, or fluorescein isothiocyanate-conjugated secondary antibody at 37 °C for 60 min. The stained sections were examined using with a Nikon confocal microscope equipped with a digital camera or by Nikon SIM.

**Co-immunoprecipitation (Co-IP)**. Co-IP was conducted using a Co-Immunoprecipitation Kit (Pierce). Spinal dorsal horn tissues were excised quickly and placed in lysis buffer. A Pierce Spin Column was placed in a microcentrifuge tube. After addition of AminoLink Plus Coupling Resin and affinity-purified transportin-1 antibody (GeneTex; GTX103003; 10 μg), the complex was incubated on a rotator at room temperature for 90–120 min to ensure antibody immobilization. Tissue lysates were added to the appropriate resin columns and incubated with gentle rocking overnight at 4 °C. The spin columns were then centrifuged and placed in new collection tubes, elution buffer was added, and the flow-through was collected by centrifugation. The immune complexes in the flow-through were analyzed by western blotting using YBX1 antibody (Abcam; ab12148; 10 μg). All co-IP steps were performed at 4 °C unless otherwise indicated.

**Dual-luciferase reporter assay**. To test the binding of miRNAs to circAnks1a or to its target gene *Vegfb*, the full-length sequence of circAnks1a or the 3'UTR of *Vegfb* was constructed into the 3' UTR of pMIR-Report Luciferase vector (Ambion). 293 T cells were co-transfected with a mixture of 200 ng pMIR-Report Luciferase vector, miRNA mimic (RiboBio) and/or pAAV-circAnks1a plasmid at a final concentration of 100 nM using Lipofectamine 2000 (Invitrogen). To investigate the effect of circAnks1a on the binding of YBX1 to the *Vegfb* promoter, the *Vegfb* promoter region containing the truncated or non-truncated putative binding area was fused to the promoterless firefly luciferase gene of the pGL3-Basic vector (Promega), and 293T cells were transfected with luciferase reporter plasmids, the pcDNA3.1-YBX1 plasmid and the pAAV-circAnks1a plasmid. After 48 h, the luciferase activity of the cells was measured using a dual-luciferase reporter assay kit (Promega). The result was normalized to the ratio between firefly activity and renilla luciferase activity.

**Western blotting**. Proteins obtained from spinal dorsal horn tissues were separated by gel electrophoresis SDS-PAGE and transferred to a PVDF membrane. The PVDF membrane was incubated with primary antibodies against VEGFB (Santa Cruz; sc-80442; 1:200), YBX1 (Abcam; ab12148; 1:1000), histone H3 (Biorbyt;

orb136531; 1:500) or β-actin (CST; 4967; 1:1500) overnight at 4 ℃. The blots were then incubated with secondary antibodies conjugated to horseradish peroxidase. The immunostained bands were quantified using a computer-assisted imaging analysis system (ImageJ). All uncropped images for full-length blots and gels were presented in the Source Data file.

**Northern blotting**. Digoxin-labeled RNA probes were prepared using the DIG Northern Starter Kit (Roche) with the corresponding PCR products as templates for T7 transcription. Total RNAs (with or without RNase R treatment) were electrophoresed on 2% agarose gels and transferred to a Hybond-N+ membrane (GE Healthcare). After crosslinking by 265-nm ultraviolet light with an energy of 200,000 mJ/cm$^2$, the membranes were pre-hybridized and hybridized with probes at 62 ℃ overnight. The membranes were then washed twice in 2× SSC, 0.1% SDS at room temperature and washed two additional times at 62 ℃. After washing, the blots were incubated with anti-digoxigenin-AP (Roche) and visualized according to the manufacturer's instructions. The sequences of the probes are shown in Supplementary Table 8b. All uncropped images for full-length blots and gels were presented in the Source Data file.

**Electrophoretic mobility shift assay (EMSA)**. EMSA was performed using a LightShift Chemiluminescent RNA EMSA Kit (Pierce) according to the manufacturer's instructions. RNA probes were labeled at the 3'-ends with biotin. Biotin-labeled RNA probes were mixed with recombinant YBX1 protein (Abcam; ab187443; 20 μg) or Transportin-1 protein (GeneTex; GTX103003; 20 μg) and incubated at room temperature for 20–30 min. Loading buffer was then added to the reaction mixture, and the sample was loaded on a native acrylamide gel. Electrophoretic separation of the RNA–protein complexes was conducted at 100 V for 60 min, and images were acquired. The sequences of the probes are shown in Supplementary Table 8a.

**Statistics**. SPSS 25.0 was used to analyze the data; the results are shown as the mean ± s.e.m. The data were analyzed using the two independent samples $t$ test or one-way ANOVA followed by Dunnett's T3 or Tukey's post hoc test. When tests of normality were not satisfied, the permutation test was substituted. The criterion of statistical significance was 0.05. Although no power analysis was performed, the sample size was determined according to previous publications in behavioral and pertinent molecular studies. All measurements were taken from distinct samples.

**Reporting summary**. Further information on research design is available in the Nature Research Reporting Summary linked to this article.

## Data availability

The RNA-seq data obtained in this study have been deposited in the SRA database, with the accession code PRJNA558403. All relevant data supporting the findings of this study are provided as a Source Data file.

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

## Acknowledgements

This study was funded by National Natural Science Foundation of China (31671090, 81771201, 81671088, 81600959, and 81801103), Natural Science Foundation of Guangdong (2016A030308003), China Postdoctoral Science Foundation (2018M631019), Science and Technology Program of Guangdong (2018B030334001), Science and Technology Program of Guangzhou (201903010047), Sun Yat-Sen University Youth Teacher Training Project (18ykpy41), and the 111 Project Grant (B13037). We thank Prof. Xu-Ri Li for providing with VEGFB$^{flox/flox}$ mice.

## Author contributions

S.Z., S.L., T.X., and W.X. designed the experiments. S.Z., S.L., M.L., C.L., H.D., and T.X. performed the experiments. Y.S. and Y.L. acquired the data. C.M., R.G., and S.W. analyzed the data. S.Z., S.L., T.X., C.M., and W.X. wrote the paper.

## Additional information

**Competing interests:** The authors declare no competing interests.

