## [Peer Review File · Nature Communications]

Reviewers' comments:

Reviewer #1 (Remarks to the Author):

The contribution of circular RNA to neuropathic pain is mostly unknown. The authors showed that the circular RNA circAnks1a was increased in the spinal cord neurons in a rat model of neuropathic pain, and induced pain-related behaviors through the increase in VEGFB expression at transcriptional and translational levels. They also reported that circAnks1a enhanced VEGFB transcription through translocation of YBX 1 into the nucleus and facilitation of YBX 1 binding to a VEGFB promoter, and enhanced VEGFB translation through sponging miR-324-3p that interacts with VEGFB mRNA.

The results are novel and interesting. My concerns are as follows.

1) The authors should show whether circAnks1a expression changes in neuropathic pain models with the primary nerve injury other than SNL to confirm that circAnks1a essentially contributes to neuropathic pain. Cao et al. reported expression changes in circRNAs in CCI models. Are there any overlapped circRNAs between the present and previous results? They also need to discuss this point.

2) The authors should show the mechanisms by which the increase in VEGFB expression caused the increase in neuronal excitability.

3) Electrophysiology is incomplete in Figs. 3 and 4. The authors only showed the changes in excitatory mEPSC, which can partly represent pre and post synaptic event. To show the increase in excitability in neurons, they should show enhanced EPSP-spike coupling and the relationship between the action potential frequency and the injected current. In addition, electrophysiological properties (e.g. resting membrane potential, input resistance and firing properties) of recorded cells should be presented.

4) Detailed information on the methods for electrophysiology is missing. There was no description on composition of ACSF, pipet internal solution, or recording solution. Further, the authors should clearly describe the cell type, the spinal layer and the spinal cord level of recorded neurons. Are the recorded cells neurons receiving direct input from the primary afferent, interneurons or the projection neurons? In case of AAV-injection experiments, how did they confirm that recorded cells were positive for AAV.

5) The authors should examine the detailed time courses of expression changes in circAnks1a, miR-324-3p and VEGFB (protein) and show whether those changes are consistent with each other in terms of time.

6) In lines 111 and 112, the authors described "circAnks1a was predominantly spinal dorsal horn in normal rat". However, they examined the expression in sham rats, not normal rats, as presented in Fig. 2F. They should show circAnks1a expression in naïve rats without any operation.

7) In Figs 2G and 2H, the authors demonstrated that circAnks1a was expressed in the nucleus and cytoplasm of dorsal horn. They should show identification of neuronal subtype and the layer and the spinal level.

8) In Fig 4, the authors showed that VEGFB was upregulated in dorsal horn neurons in circAnks1a-injected and SNL rats. To confirm that VEGFB expression was regulated by circAnks1a, they should show that the increase in both molecules occurs in the identical neurons. Although they showed that YBX1 expression was colocalized with circAnks1a in Fig. 6D and VEGF-positive cells in Fig. 7A, respectively, both photos do not seem to be persuasive.

9) In Fig.6, the authors presented increased YBX1 expression in the nucleus in SNL rats, and explained that the increased circAnks1a translocates YBX1 into the nucleus. Is YBX1 decreased in the cytoplasm, and is total amount of YBX1 unchanged?

10) In Fig. 8G, the authors showed that SNL treatment enhanced binding of circAnks1a onto the VEGFB promoter. How does circAnks1a binds to the Vegf promoter? What mechanisms work for the enhancement?

11) In Fig.10J, miR-324-3p antagomir induced the pain-related behavior in naïve rats. Given that broad targeted mRNAs for miR-324-3p, is there a possibility that targeted mRNAs other than VEGFB, e.g. Kv4.2, are responsible for the induction of hyperalgesia and mechanical allodynia in this experiment?

- 12) In Supplementary Table 1, when were sham (control) samples collected, day 7 or day 14? Please clearly state it.
- 13) Please describe the detail of PhyloP database.
- 14) More detailed information on AAV virus used needs to be provided, including the serotype and the construction methods.
- 15) Detailed information on the methods for RNA pulldown assay, ChIP, ChIRP and IP is not presented. The sequence of circAnks1a seems to be incorrect. Please check it.
- 16) In the Discussion, the authors need to describe whether miR-324-3p targets Vegfb mRNA in human.

Reviewer #2 (Remarks to the Author):

This is a deep investigation of circRNA in neuropathic pain. The authors have provided evidence of regulatory roles for the circRNA by performing a large number of experiments in SNL animal model. They have discovered differential expression of circAnks1a and a suite of mechanisms by which circAnks1a regulates VEGFB in dorsal horn neurons following nerve injury. Specifically, the authors demonstrate that circAnks1a is capable of recruiting transcription factor YBX1 for nuclear translocation and association with the VEGFB promoter. They also show that this induces VEGFB transcription and increases the perception of pain in vivo. circAnks1a was also found to be capable of sponging miR-324-3p in the cytoplasm to facilitate de-repression of posttranscriptional VEGFB mRNA regulation by the miRNA and increase its target mRNA translation. The study is extensive and seems well conducted with the figures and tables presenting the results appropriately. Nevertheless, there is some revisions that the authors might consider to improve the manuscript:

Major revisions:

- The main concern about the manuscript is that there is two different mechanisms of VEGFB regulation by circAnks1a are suggested, including YBX1 recruitment and miR-324-3p inhibition, however the this would indicate a surprising degree of redundancy, particularly given that these two mechanisms regulate a single gene in different compartments of the cell (transcription in nucleus and sponging in cytoplasm). Also, the degree of contribution of each mechanism remains unclear- do the two mechanisms work in alternate or simultaneously? In either case, how are these controlled? Is anything known about the specificity of back splicing for circAnks1a?
- I am slightly confused about how circAnks1a mediates YXB1 binding at the VEGFB promoter? What is the nature of the interaction? Presumably it is Watson-Crick base pairing, but it would be nice if this could be illustrated in a diagram showing putative secondary structure. The observation that circAnks1a-VEGFB promoter interaction by ChIRP was supported by in silico analysis by IntaRNA, LucRNA and LncTar should include more information about this data and what it means. Was the YXB1 binding site in circAnks1a a Y box consensus sequence?
- It was shown that circAnks1a upregulation altered 29 genes including VEGFB, but it was not mentioned if the authors detected a change in the YBX1? If not, it would be against the suggested recruitment mechanism that involves all three molecules, circAnks1a, YBX1 and VEGFB. Could any of these other genes alter signal transduction and modulate the perception of pain? Can the authors suggest a mechanism for altering the expression of these other genes through direct or indirect influence of circAnks1a changes?
- The authors also identified several other circRNA differentially expressed in the SNL model but we didn't hear anything about these? Are any of the molecules capable of binding miRNA associated with the model? Have any of these molecules been found to be altered previously in other biological systems? Are the host genes associated with any biological system or function?

- It wasn't clear how the recombinant AAV-hSyn-circAnks1a-hSyn-EGFP works? A plasmid map and diagram showing how circAnks1a is processed from the pri-mRNA transcript produced from this construct would be helpful. How are the exons arranged induce back splicing or is some other mechanism used to circularize the transcript. Is the product identical to the endogenous circAnks1a?

- Given that the authors used RNAseq to identify circRNA after RNase R depletion of linear RNA transcripts, how were the read counts normalized as this method can have variable impact on the background reads used to normalize RNA sequencing? Did the authors quantify the circRNA enrichment achieved by nuclease digestion?

- What was the relative amounts/expression level of circAnks1a in the dorsal horn tissue compared to other circRNA and other translated linear RNA, particularly linear mRNA for Anks1a? What is the function of the protein coding Anks1a. Does it have any function in sensory neurons?

- The authors performed dual-luciferase assay and showed that circAnks1a promoted the transcription of VEGFB by YBX1 via position -1687 to -805 1834. However, another control without the YBX1 was required to confirm that the observed consequence was not due to the circAnks1a-VEGFB interaction, but rather resulted from both circAnks1a and YBX1.

- In section "circAnks1 acts as mir324-3p competitor" the authors justify that because the YBX1 knock down completely suppressed the increase of VEGFB mRNA but only partially reduced the increased protein, then this may be due to a posttranscriptional (miRNA) regulation. However, this seems unreasonable because less reduction of the protein compared to mRNA does not imply miRNA regulation, rather the opposite is usually the case.

Minor revisions:

- Line 86, "by criteria of at least one unique back-spliced reads" does not make sense thus needs to be reworded.

- Line 93, perhaps change "unregulated" to "dysregulated".

- Line 402, "translational" should be replaced by "posttranscriptional".

- SNL acronym is in the title and not defined till the methods.

Reviewer #3 (Remarks to the Author):

In this manuscript, Zhang et al profiled the change of circRNA abundance in rat spinal dorsal horn following SNL (I guess it means spinal nerve injury?) treatment and identified a circRNA derived from Anks1a gene locus (circAnks1a) with the most significant upregulation. The authors then went on to show the increased expression of this circRNA could indeed enhance the excitation of dorsal horn neuron and mediated the neuropathic pain behavior via both knockdown and overexpression experiments. Moreover, by comparing the mRNA expression level in dorsal horns with circAnks1a overexpression to those without, they identified VEGFB as the potential target and further demonstrated that VEGFB could mediated the effect induced by SNL as well as circAnks1a overexpression. Finally, the authors investigated the molecular mechanisms underlying the regulation of VEGFB by circAnks1a. They found that the circRNA on one hand could induce VEGFB transcription by enhancing a transcription factor YBX1 nuclear localization as well as its binding at VEGFB promoter, on the other hand could sponge the mir-324-3p, thereby alleviate its suppression of VEGFB mRNA translation. Although it appeared to uncover a new circRNA with important role in SNL and its underlying regulatory mechanism, the current study lacks many

technical details and enough controls, without which it is difficult to judge the results.

Major comments:

1. The authors profiled the abundance of circRNA based on ribo- RNA-seq. It is not clear what computational tool they used for identifying circRNA and estimating their abundance. They claimed only one unique junction read was required as evidence of circRNA expression. Did they estimate the false positive rate based on such low criteria? Also it is not clear whether they required the back-splicing junctions cover the known splicing sites. If so, how can some circRNAs derive solely from intronic sequences? If not, what would be the biogenesis mechanisms for those not covering splicing sites.
2. The whole study was focused on one circRNA, circAnks1a. It is therefore important to validate its circular feature and its full-length sequences. The golden standard for circRNA validation is Northern Blot, with and without RNase R treatment, with and without linearization.
3. The authors did not show the specificity of siRNA knockdown. Did it also affect linear mRNA?
4. The authors did not describe the design of circRNA overexpression constructs and estimated the efficiency of its circularization.
5. The FISH experiments was done without enough control. The authors should use probes targeting at the sequences shared by linear and circular Anks1a, as well as those targeting only at linear Anks1a, as control.
6. The authors claimed that circAnks1a could enhance the binding between YBX1 and transportin-1. Whereas they showed the binding of YBX1 and circAnks1a, they did not test the interaction between circAnks1a and transportin-1. They also should identify the exact binding sites of both YBX1 and transportin-1, and then compare the effect between overexpression of wildtype circAnks1a and circAnks1a with the binding sites mutated.
7. The authors claimed that circAnks1a could enhance the binding of YBX1 at VEGFB promoter. However, they did not provide the direct evidence, i.e. compare the binding of YBX1 at VEGFB promoter with to without circAnks1a overexpression by ChIP.
8. The authors claimed that circAnks1a could affect VEGFB translation via sponging the repressive mir-324-3p. One essential control experiment would be to compare the effect between overexpression of wildtype circAnks1a and circAnks1a with the potential mir-324-3p binding sites mutated.
9. Linear Anks1a mRNA covered all the circAnks1a sequence except the junction. Could linear Anks1a have the same effect? The authors should also estimate the relative abundance between its linear and circular isoforms in different conditions.
10. Finally, the authors have applied quite a few technically challenging methods, such as ChIRP, RNA pull-down assays, and etc. However, there is no detailed description in the Method part. It is therefore impossible to judge the quality of their data.

Reviewer's comments:

Reviewer#1

Comments 1: The authors should show whether circAnks1a expression changes in neuropathic pain models with the primary nerve injury other than SNL to confirm that circAnks1a essentially contributes to neuropathic pain. Cao et al. reported expression changes in circRNAs in CCI models. Are there any overlapped circRNAs between the present and previous results? They also need to discuss this point.

Responses: We sincerely appreciate the reviewer's comments. Following the reviewer's suggestion, we examined the circAnks1a expression in dorsal horn on day 7 and day 14 in rats with neuropathic pain induced by sciatic chronic constriction injuries (CCI). The PCR results showed a significantly increased expression of circAnks1a in dorsal horn in rats with CCI, when compared with those in the sham group (Comment Fig. 1), which implied the essential involvement of circAnks1a in the neuropathic pain.

In addition, we compared the differential expression of circRNAs between Cao's and our data. We found that the circRnf4 (circ:chr14:81667646-81672500) was upregulated in high throughput data from both studies. However, further PCR results showed no significant difference of circRnf4 expression between sham and SNL group in the present study (Supplementary Table 3), while circAnks1a did not show the differential expression in Cao's data. The discrepancy may result from various compounding factors including the different experimental conditions and different detection methods (RNA sequencing versus RNA microarray). We added additional

comments to address this issue in Discuss section in the revised manuscript.

Comment Fig. 1

Comments 2: The authors should show the mechanisms by which the increase in VEGFB expression caused the increase in neuronal excitability.

Responses: VEGFB, as a member of the VEGF family, can only bind and activate tyrosine kinase receptors named as vascular endothelial growth factor receptor 1 (VEGFR1). Studies showed that VEGFA, VEGFB and PLGF can bind VEGFR1 with distinct functions. VEGFB binding leads to the activation of a number of downstream activators similar to most tyrosine kinase receptors, including p38 MAPK, ERK/MAPK, PKB/AKT and PI3K^{1,2}. It is well known that activation of p38 MAPK or PKB/AKT can initiate inflammation response in the nervous system. Our pilot experiments have demonstrated that intraperitoneal injection of ERK inhibitor PD98059 alleviates the increase of mEPSC frequency and amplitude in dorsal horn neurons (Comment Fig. 2A) and mechanical allodynia induced by the AAV-VEGFB-EGFP intraspinal injection (Comment Fig. 2B). An ongoing study in our lab is well on the way to explore the intracellular process underlying VEGFB-mediated neuronal hyperexcitability and allodynia in the rodent models with neuropathic pain. These results raised the possibility that the activation of tyrosine kinase (e.g., ERK signaling) might contributed to the VEGFB-mediated the neuronal excitability increase.

Comment Fig. 2

Comments 3: Electrophysiology is incomplete in Figs. 3 and 4. The authors only showed the changes in excitatory mEPSC, which can partly represent pre and post synaptic event. To show the increase in excitability in neurons, they should show enhanced EPSP-spike coupling and the relationship between the action potential frequency and the injected current. In addition, electrophysiological properties (e.g. resting membrane potential, input resistance and firing properties) of recorded cells should be presented.

Responses: We appreciate the comments from the reviewer. To address these comments, additional electrophysiological studies were performed. The results showed that the number of C-fiber–evoked action potentials was increased on day 14 after SNL in rats, which was significantly attenuated by circAnks1a siRNA (Comment Fig. 3A). Furthermore, significantly increased depolarization-induced neuronal firing frequency (Comment Fig. 3B) and EPSC-spike coupling (Comment Fig. 3C) were observed in dorsal horn neurons in the rats with SNL, which were also attenuated by intrathecal injection of circAnks1a siRNA (Comment Fig. 3B and 3C). We also observed the significant positive shift of resting membrane potential in dorsal horn neurons on day 14 in the modeled rats, which was recovered by circAnks1a siRNA (Comment Fig. 3D). We also noted that intrathecal injection of mAnks1a siRNA failed to ameliorate these neuronal dysfunctions in dorsal horn neurons in the modeled rats. We added these findings into the revised manuscript.

Comment Fig. 3

Comments 4: Detailed information on the methods for electrophysiology is missing. There was no description on composition of ACSF, pipet internal solution, or recording solution. Further, the authors should clearly describe the cell type, the spinal layer and the spinal cord level of recorded neurons. Are the recorded cells neurons receiving direct input from the primary afferent, interneurons or the projection neurons? In case of AAV-injection experiments, how did they confirm that recorded cells were positive for AAV.

Responses: According to the reviewer's suggestion, we provided the detailed information about the composition of ACSF, pipet internal solution and recording solution in the methods for electrophysiology.

Dorsal horn lamina I-II neurons from the L4-L6 were chosen for electrophysiological recording. Immunostaining results were also presented to show that the recorded biocytin-positive cells were colocalized with the NK1R-positive immunoreactivity (Comment Fig. 4), suggesting that the recorded neurons are NK1R+ projection neurons of spinal superficial lamina. We added the description into the revised manuscript.

Comment Fig. 4

Due to the great diversity of dorsal horn neurons and the complex circuit, it is difficult

to determine the exact input source of the recorded cells. In the present study, the recorded neurons are NK1R-positive cells, and it is generally recognized that 80% NK1R-positive cells in dorsal horn are projection neurons with primary afferent input^{3, 4}. So, it is likely that the majority of recorded cells received the input from the primary afferents.

In the present study, the recorded EGFP-positive cells indicated the AAV-infected cells (Comment Fig. 5).

Comment Fig. 5

Comments 5: The authors should examine the detailed time courses of expression changes in circAnks1a, miR-324-3p and VEGFB (protein) and show whether those changes are consistent with each other in terms of time.

Responses: According to the reviewer’s suggestion, we examined the expression changes in circAnks1a, miR-324-3p and VEGFB on sham and day 3, 7, 10 and 14 following SNL. As shown below, SNL significantly increased the expression of circAnks1a (Comment Fig. 6A) and VEGFB (Comment Fig. 6B), and the time course of circAnks1a upregulation was consistent with that of VEGFB. However, the expression of miR-324-3p did not show an obvious change on sham and day 3, 7, 10 and 14 following SNL (Comment Fig. 6C). We added these results into Fig. 1d and 4d in the revised manuscript.

Comment Fig. 6

Comments 6: In lines 111 and 112, the authors described “circAnks1a was predominantly spinal dorsal horn in normal rat”. However, they examined the expression in sham rats, not normal rats, as presented in Fig. 2F. They should show circAnks1a expression in naïve rats without any operation.

Responses: we greatly appreciated the comments. Here, we observed the circAnks1a expression in dorsal horn in the naïve animals. In the present study, there is no difference in the expression of circAnks1a between the naïve group and the sham group (Comment Fig. 7). We added this new graph into Fig. 2g in the revised manuscript.

Comment Fig. 7

Comments 7: In Figs 2G and 2H, the authors demonstrated that circAnks1a was expressed in the nucleus and cytoplasm of dorsal horn. They should show identification of neuronal subtype and the layer and the spinal level.

Responses: According to the reviewer’s suggestion, we examined the neuronal subtype and the layer where circAnks1a was expressed. We found that the immunoreactivity of circAnks1a was colocalized with the NK1R-positive cells (Comment Fig. 8A and B). Furthermore, the expression of circAnks1a was detected in different level of spinal from T13 to S3, and the highest expression is in the lamina I-III of spinal L4-L6 (Comment Fig. 8C). We have added these results into Supplementary Fig. 1a and 2d.

Comment Fig. 8

Comments 8: In Fig 4, the authors showed that VEGFB was upregulated in dorsal horn neurons in circAnks1a-injected and SNL rats. To confirm that VEGFB expression was regulated by circAnks1a, they should show that the increase in both molecules occurs in the identical neurons. Although they showed that YBX1 expression was colocalized with circAnks1a in Fig. 6D and VEGF-positive cells in Fig. 7A, respectively, both photos do not seem to be persuasive.

Responses: According to the reviewer's suggestion, we performed the FISH staining using circAnks1a probe and VEGFB antibody. The results showed that the increase in circAnks1a and VEGFB occurs in the same neurons in spinal dorsal horn following SNL (Comment Fig. 9). We added this result into the Fig. 5a in the revised manuscript.

Comment Fig. 9

Comments 9: In Fig.6, the authors presented increased YBX1 expression in the nucleus in SNL rats, and explained that the increased circAnks1a translocates YBX1 into the nucleus. Is YBX1 decreased in the cytoplasm, and is total amount of YBX1

unchanged?

Responses: According to the reviewer's suggestion, we observed the expression of YBX1 in the total and cytoplasm, respectively. The results showed that the total amount of YBX1 did not change, but the cytoplasm YBX1 level was significantly decreased, on day 7 and 14 following SNL (Comment Fig. 10). We added these results into the revised manuscript in Supplementary Fig. 6b and 6c.

Comment Fig. 10

Comments 10: In Fig. 8G, the authors showed that SNL treatment enhanced binding of circAnks1a onto the VEGFB promoter. How does circAnks1a binds to the Vegf promoter? What mechanisms work for the enhancement?

Responses: We greatly appreciated the comments of reviewer. In the present study, we used three bioinformatics softwares to analyze the potential binding sites between circAnks1a and VEGFB promoter. LncRNA (LongTarget) was developed to predict ncRNA's DNA binding motifs and binding sites in a genomic region based on potential base pairing rules. LncRNA (LongTarget) analysis showed that the region of VEGFB promoter (-1510~-1451) was a potential binding site for circAnks1a. Further analysis with IntaRNA, which is a program for the accurate prediction of interactions between two RNA molecules, indicated that a specific sequence in circAnks1a (from 228 to 375) may be a high affinity site for the region of -1834 to -1687 of *Vegfb* promoter (energy = -24.08 kcal/mol). Meanwhile, LncTar analysis showed that circAnks1a could bind with the region of -2000 to -1041 of *Vegfb* promoter (ndG = -0.0605). Based on these databases analysis, we performed the chromatin isolation by

RNA purification (ChIRP) assay *in vitro*, and found that circAnks1a markedly bound to *Vegfb* promoter at position -1834 to -1687, but not -1510 to -1451, in circAnks1a-overexpressed C6 cells. With *in vivo* experiments, we further confirmed that SNL treatment significantly enhanced the binding level of circAnks1a on *Vegfb* promoter at the position -1834 to -1687 compared with sham group. A diagram about the binding between circAnks1a and the *Vegfb* promoter was shown below (Comment Fig. 11). We added this diagram into Fig. 8h in the revised manuscript. In addition, peer's study showed that noncoding RNA such as LncRNA-BX111 promoted the binding of transcriptional factor to ZEB1 promoter⁵. In the present study, the increased circAnks1a in nucleus facilitated the interaction of YBX1 and *Vegfb* promoter, and enhanced the VEGFB expression, compared with the YBX1 *per se*.

Comment Fig. 11

Comments 11: In Fig.10J, miR-324-3p antagomir induced the pain-related behavior in naïve rats. Given that broad targeted mRNAs for miR-324-3p, is there a possibility that targeted mRNAs other than VEGFB, e.g. Kv4.2, are responsible for the induction of hyperalgesia and mechanical allodynia in this experiment?

Responses: We appreciated the comments from reviewer greatly. According to the reviewer's suggestion, we examined whether miR-324-3p can regulate the Kv4.2 expression, and found that intrathecal injection of miR-324-3p agomir did not rescue the decreased Kv4.2 expression induced by SNL (Comment Fig. 12A). Furthermore, intrathecal injection of VEGFB siRNA recused the mechanical allodynia induced by miR-324-3p antagomir (Comment Fig. 12B). These results suggested that VEGFB, but not other molecular such as Kv4.2, is a key target of miR-324-3p in chronic pain induced by SNL.

Comment Fig. 12

Comments 12: In Supplementary Table 1, when were sham (control) samples collected, day 7 or day 14? Please clearly state it.

Responses: In the present study, the sham (control) samples were collected on day 14. We have stated it in the revised manuscript.

Comments 13: Please describe the detail of PhyloP database.

Responses: PhyloP is a program in Phylogenetic Analysis with Space/Time models (PHAST) software package. PHAST consists of a collection of programs and supporting libraries for statistical phylogenetic modeling and functional element identification, which is best known as the engine behind the conservation tracks in the University of California, Santa Cruz (UCSC) Genome Browser⁶. PhyloP is a program aiming to compute of P-values for conservation of either lineage specific or across all branches⁷. We downloaded a directory containing conservation scoring by phyloP for multiple alignments of 19 genomes to the rat genome from UCSC⁸. We then read out the conservation scores along the full length of circRNA and searched for blocks of at least 6-nucleotide length that exceeded a conservation score of 0.5. These blocks are counted as conservative blocks. We added further details in the revision.

Comments 14: More detailed information on AAV virus used needs to be provided,

including the serotype and the construction methods.

Responses: In the present study, the serotype of AAV2/9 was used. AAV virus were all designed and constructed by standard methods with the help from Obio Technology (Shanghai) or BrainVTA (Wuhan). Briefly, a modified vector pAAV-hSyn-polyA was obtained by replacing the DNA fragment between hSyn and right ITR with PCR amplified polyA and the flanking cloning sites. The sequence of target gene was synthesized and inserted following hSyn. AAV-293 cells were transfected with pAAV-RC, pHelper together with pAAV vector. 48 hours after transfection, virus was purified from the cell lysate by using a heparin-agarose column and concentrated to the final volume. The virus titer was measured using quantity PCR method. According to the reviewer's suggestion, we added these detailed descriptions on AAV virus into the *methods* of revised manuscript.

Comments 15: Detailed information on the methods for RNA pulldown assay, ChIP, ChIRP and IP is not presented. The sequence of circAnks1a seems to be incorrect. Please check it.

Responses: We added more technical details for RNA pulldown assay, ChIP, ChIRP and IP in the methods of revised manuscript. Furthermore, we have corrected the typing error about the sequence of circAnks1a.

Comments 16: In the Discussion, the authors need to describe whether miR-324-3p targets Vegfb mRNA in human.

Responses: By the bioinformation analysis (miRanda), we found that miR-324-3p can target Vegfb mRNA in human as shown in the following analog diagram (Comment Fig. 13). According to the reviewer's suggestion, we added some discussion in the revised manuscript.

Comment Fig. 13

Reviewer #2

Comments 1: The main concern about the manuscript is that there is two different mechanisms of VEGFB regulation by circAnks1a are suggested, including YBX1 recruitment and miR-324-3p inhibition, however the this would indicate a surprising degree of redundancy, particularly given that these two mechanisms regulate a single gene in different compartments of the cell (transcription in nucleus and sponging in cytoplasm). Also, the degree of contribution of each mechanism remains unclear- do the two mechanisms work in alternate or simultaneously? In either case, how are these controlled? Is anything known about the specificity of back splicing for circAnks1a?

Responses: We appreciated the great comments from the reviewer. The present study illustrated two potential circAnks1a-mediated mechanism underlying VEGFB upregulation in the rodent model with nerve injury. In the present study, suppression of circAnks1a by specific siRNA completely reduced the VEGFB upregulation induced by nerve injury, while YBX1 siRNA only partially mitigated VEGFB protein upregulation in the same setting. This implied that both mechanisms, YBX-1 recruitment and miR-324-3p inhibition, might simultaneously regulate *Vegfb* transcription and mRNA translation in the modeled rodents, while we cannot accurately quantitate the accurate weight of each mechanism in the process of VEGFB upregulation, due to the potential intrinsic compensation and interaction between two mechanisms, in the rodents with nerve injury. Various mechanisms, including (but not limited to) cytoplasmic/nuclear shuttling after circRNA biogenesis, the functional status of other transcriptional regulators (e.g., transcriptional factors and the interaction between potential circRNA and transcriptional factors), and the regulation of miRNA abundance and its action with circRNA, may potentially regulate the circAnks1a-involved pathway to induce VEGFB upregulation in the setting of neuropathic pain. Emerging studies to address similar questions are presented in the frontline of circRNA biology^{9,10}. We added additional comments to address this limitation in Discussion section in the revision.

To validate the specificity of back splicing for circAnks1a, we first examined the host Anks1a gene-derived all circRNA, which included circAnks1a (circ:chr20:7561057-7573740) and circAnks1a.2 (circ:chr20:7562320-7573740). The PCR results showed that the abundance of circAnks1a was higher than that of circAnks1a.2, and the level of circAnks1a, but not circAnks1a.2, was significantly

increased following SNL. These results are consistent with the peer's opinion that there often exists a predominantly expressed circRNA isoform from one gene locus¹¹, and indicated that the present circAnks1a are specifically expressed in both physiological and pain states (Comment Fig. 14).

Comment Fig. 14

In addition, we also validated the specificity of circAnks1a using Northern Blot. The results showed that circAnks1a can be detected with and without RNase R treatment using circAnks1a probe. When detecting with mAnks1a probe, the mAnks1a can only be detected without RNase R (Comment Fig. 15). The results confirmed the specificity of back splicing of circAnks1a.

Comment Fig. 15

Comments 2: I am slightly confused about how circAnks1a mediates YXB1 binding at the VEGFB promoter? What is the nature of the interaction? Presumably it is Watson-Crick base pairing, but it would be nice if this could be illustrated in a diagram showing putative secondary structure. The observation that circAnks1a-VEGFB promoter interaction by ChIRP was supported by in silico

analysis by IntaRNA, LncRNA and LncTar should include more information about this data and what it means. Was the YXB1 binding site in circAnks1a a Y box consensus sequence?

Responses: We greatly appreciated the comments of reviewer. The nature of the interaction between circAnks1a and Vegfb promoter, based on the database analysis, is Watson-Crick base pairing. In addition, bioinformatics analysis (ATTRACT, Genomatix) and peer's studies showed that YBX1, as a RNA binding protein, has been reported to bind the non-coding RNA such as circRNA or LncRNA^{5, 9}. According to the reviewer's suggestion, a diagram was presented to illustrate the interaction of circAnks1a, YBX1 and Vegfb (Comment Fig. 16). We added the diagram into Fig. 8h in the revised manuscript.

Comment Fig. 16

In the present study, we used three bioinformatics software to analysis the potential binding site between circAnks1a and Vegfb promoter. LncRNA (LongTarget) was developed to predict ncRNA's DNA binding motifs and binding sites in a genomic region based on potential base pairing rules, and the calculating results showed that region of Vegfb promoter (-1510~-1451) was a potential binding site for circAnks1a. Furthermore, the analysis results of IntaRNA, which is a program for the accurate prediction of interactions between two RNA molecules, indicated that circAnks1a (from 228 to 375) may be a site with high affinity site for the region of -1834 to -1687 of Vegfb promoter (energy= -24.08kcal/mol). At last, LncTar calculating showed that circAnks1a could bind with the region of -2000 to -1041 of Vegfb promoter (ndG=-0.0605). According to the reviewer's suggestion, we added these descriptions into the *material and methods* in the revised manuscript.

In the present study, the YBX1 binding site in circAnks1a was not a Y box consensus sequence.

Comments 3: It was shown that circAnks1a upregulation altered 29 genes including

VEGFB, but it was not mentioned if the authors detected a change in the YBX1? If not, it would be against the suggested recruitment mechanism that involves all three molecules, circAnks1a, YBX1 and VEGFB. Could any of these other genes alter signal transduction and modulate the perception of pain? Can the authors suggest a mechanism for altering the expression of these other genes through direct or indirect influence of circAnks1a changes?

Responses: We appreciated the great comments. In the circAnks1a overexpression experiment, the increase of circAnks1a induced the YBX1 translocated into nucleus (Fig. 6h in manuscript) to facilitate the VEGFB expression, but not the expression of total YBX1 (Comment Fig. 17). We added the result into the revised manuscript in Supplementary Fig. 6d.

Comment Fig. 17

In the present study, we found that several mRNA including Heph (Hephaestin), Piezo2, Plac8 (Placenta specific 8), Fap (fibroblast activation protein) and Vegfb was significantly increased in the dorsal horn following AAV2/9-circAnks1a-EGFP injection. To date, no report to show that Heph, Plac8 or Fap is involved in the neuropathic pain. The present study demonstrated the circAnks1a potentially regulated mRNA expression through direct or indirect pathway. For example, we reported that circAnks1a may partner with YBX1 to facilitate the Vegfb transcription and directly increases Vegfb mRNA, and it also sponges miR-324-3p, thus indirectly stabilizing Vegfb mRNA and increasing its content in dorsal horn following nerve

injury. The mechanism underlying circAnks1a-induced mRNA upregulation may vary, depending on specific gene sequence and mRNA-specific regulatory mechanism.

Comments 4: The authors also identified several other circRNA differentially expressed in the SNL model but we didn't hear anything about these? Are any of the molecules capable of binding miRNA associated with the model? Have any of these molecules been found to be altered previously in other biological systems? Are the host genes associated with any biological system or function?

Responses: According to the reviewer's suggestion, we first listed the gene names of the corresponding up-regulated circRNA, and retrieved all upregulated circRNA on Pubmed. Three circRNAs including circSNX29¹² and circCREBBP¹³ were studied by peers. In addition, we predicted the potential binding miRNA and retrieved the function of their host genes. We summarized major information in the below table (Comment Table 1).

Information summary of differentially expressed circRNAs

CircRNA	Predicted miRNA	Gene	CircRNA function	Gene Function
circ:chr10:4140144-4172981	rno-miR-1956-3p	Snx29	Proliferation/Differentiation	Microtubule motor activity
circ:chr3:8747385-8750638	rno-miR-326-3p rno-miR-132-5p rno-miR-320-5p rno-miR-3557-3p	LOC499770	/	/
circ:chr1:16514097-16530121	rno-miR-370-3p* rno-miR-3575	Ahi1	/	Leukemia and brain disorders
circ:chr10:11669353-11670969	rno-miR-3587	Crebbp	Primary breast cancers	Female sexual behavior
circ:chr2:172320327-172320627	rno-miR-181b-2-3p rno-miR-653-3p rno-miR-1949 rno-miR-3595	Schip1	/	Regulator of Hippo signaling
circ:chr9:100507197-100507524	rno-miR-301a-5p rno-miR-489-3p rno-miR-741-3p	/	/	/
circ:chr1:250833012-250919167	rno-miR-3557-3p rno-miR-6216	Sgms1	/	Brain disorders
circ:chr14:81667646-81672500	rno-miR-28-3p rno-miR-743b-5p rno-miR-708-3p	Rnf4	/	Targeting a ubiquitin ligase
circ:chr6:23270238-23279721	rno-miR-382-3p	Clip4	/	Gastric cancer
circ:chr17:23684383-23701761	rno-miR-667-5p	Phactr1	/	Proliferation/Migration
circ:chr8:93948882-94008016	rno-miR-370-3p* rno-miR-667-5p	Ube3d	/	Macular degeneration
circ:chr12:19311511-19312478	rno-miR-30e-5p rno-miR-30d-5p rno-miR-30a-5p	Mcm7	/	Oncogenic activity
circ:chr10:109122252-109148654	rno-miR-212-5p rno-miR-485-5p rno-miR-672-5p	Baiap2	/	Brain disorders
circ:chr2:188886906-188931941	rno-miR-218a-1-3p	Kcnn3	/	Afterhyperpolarization
circ:chr18:27956412-27984766	rno-miR-30e-3p rno-miR-30a-3p rno-miR-666-3p	Ctnna1	/	Gastric cancer
circ:chr4:84682374-84706882	rno-miR-107-5p rno-miR-136-3p rno-miR-291b	Scrn1	/	Colorectal cancer

circ:chr3:122130941-122141403	rno-miR-29b-5p rno-miR-125a-3p* rno-miR-149-5p	Sirpa	/	Tyrosine kinase receptor pathway
circ:chr2:172312783-172350004	rno-miR-3542	Schip1	/	Cytoskeleton Rearrangements
circ:chr1:79848478-79856028	rno-miR-328a-5p rno-miR-466b-5p rno-miR-466d	/	/	/
circ:chr7:11216020-11216322	rno-miR-345-3p rno-miR-17-1-3p rno-miR-143-5p rno-miR-196a-5p	LOC690617	/	/

*indicating miRNAs possibly associated with pain

Comment Table. 1

Comments 5: It wasn't clear how the recombinant AAV-hSyn-circAnks1a-hSyn-EGFP works? A plasmid map and diagram showing how circAnks1a is processed from the pri-mRNA transcript produced from this construct would be helpful. How are the exons arranged induce back splicing or is some other mechanism used to circularize the transcript. Is the product identical to the endogenous circAnks1a?

Responses: In the present study, virus was AAV-hSyn-circAnks1a-nEF1 α -EGFP but not AAV-hSyn-circAnks1a-hSyn-EGFP. We corrected the writing error in the revised manuscript.

The vector was constructed mainly according to Dongming Liang's method to facilitate circularization of circRNA¹⁴. We constructed the full-length sequence of circAnks1a into a modified vector pAAV-hSyn-polyA. Importantly, a pair of short inverted repeats (CR1 and CR1 RC) was inserted flanking the full-length sequence of circAnks1a. The short inverted repeats (36 nt) will base-pair to one another, thereby bringing the splice sites into close proximity to each other. The sequence of CR1 and CR1 RC were cgctgtcggataatgtgggcacagccgagccgtgtt and aacacggctcggctgtgccccacattatccgacagcg, respectively. This method had been proven to allow the intervening exons to efficiently circularize. The plasmid map and diagram are shown below (Comment Fig. 18). We transfected the plasmid into 293T cells and collected cellular RNA after 48 hours. Sanger sequencing showed that the full-length sequence of circAnks1a from plasmid was identified to the endogenous circAnks1a. We added these descriptions in the methods of revised manuscript.

Comment Fig. 18

Comments 6: Given that the authors used RNAseq to identify circRNA after RNase R depletion of linear RNA transcripts, how were the read counts normalized as this method can have variable impact on the background reads used to normalize RNA sequencing? Did the authors quantify the circRNA enrichment achieved by nuclease digestion?

Responses: We greatly appreciated the comment from reviewer. While previous studies showed that circRNA can be identified using RNA seq with^{15, 16} or without¹¹ RNase, evidence existed to demonstrate that RNase R treatment depleted the samples from linear RNAs and increased the abundance and accuracy of circRNA identification^{17, 18}. We then adopted the method with RNase treatment for sequencing. We recognized that this method can have variable impact on the background reads, so we calculate RPM (Reads Per Million mapped reads) to normalize and quantify each circRNA as mentioned before^{15, 18}. Following the comments from reviewer, we compared the content of circAnks1a and mAnks1a between the lysates with RNase R and that without RNase R. We found that application of RNase R significantly decreased the mAnks1a level, but did not affect the circAnks1a level (Comment Fig. 19)

Comment Fig. 19

Comments 7: What was the relative amounts/expression level of circAnks1a in the dorsal horn tissue compared to other circRNA and other translated linear RNA, particularly linear mRNA for Anks1a? What is the function of the protein coding Anks1a. Does it have any function in sensory neurons?

Responses: According to the reviewer's suggestion, we measured the relative amounts of circAnks1a compared to other circRNAs and the linear Anks1a mRNA in the dorsal horn tissue in the sham group and SNL group. We found a relative substantial abundance of circAnks1a (red font) compared with the other circRNA or linear Anks1a, in sham group. We also found that SNL induced greatest upregulation of circAnks1a (Comment Fig. 20). Importantly, SNL significantly increased the level of circAnks1a RNA, but not linear Anks1a mRNA in the dorsal horn.

Comment Fig. 20

Anks1a is widely expressed, and comprised of six aminoterminal Ankyrin motifs followed two sterile motifs (SAM) and a phosphotyrosine binding domain (PTB). The role of Anks1a has been mainly investigated in the signaling pathway downstream of the epidermal growth factor receptor and Eph receptor^{19,20}. To date, no data showed that Anks1a protein have any function in sensory neurons. Here, we found no significant change of Anks1a expression in the rodents with SNL, and suppression of Anks1a by siRNA failed to modify pain sensitivity in the rodents with nerve injury (Fig. 3d and 3e in the manuscript).

Comments 8: The authors performed dual-luciferase assay and showed that circAnks1a promoted the transcription of VEGFB by YBX1 via position -1687 to -805 1834. However, another control without the YBX1 was required to confirm that the observed consequence was not due to the circAnks1a-VEGFB interaction, but rather resulted from both circAnks1a and YBX1.

Responses: We greatly appreciated the reviewer's comment. According to the reviewer's suggestion, we performed additional studies with the dual-luciferase assay. The results showed that the luciferases activity is higher in VEGFB+YBX1 group compared with the non-YBX1 group (Comment Fig. 21). The results further confirmed that VEGFB upregulation was not due to the circAnks1a-VEGFB interaction, but rather resulted from both circAnks1a and YBX1. We added the results into the revised manuscript in Fig. 8c.

Comment Fig. 21

Comments 9: In section “circAnks1 acts as mir324-3p competitor” the authors justify that because the YBX1 knock down completely suppressed the increase of VEGFB mRNA but only partially reduced the increased protein, then this may be due to a posttranscriptional (miRNA) regulation. However, this seems unreasonable because less reduction of the protein compared to mRNA does not imply miRNA regulation, rather the opposite is usually the case.

Responses: We greatly appreciated the reviewer’s comments. Generally, we agreed the comment from the reviewer regarding the differential expression of mRNA and protein indicating the potential miRNA-involved regulation of protein expression, when we discussed the regulatory effect of a single mediator on mRNA translation. Here, we reported that YBX1, partnering with circAnks1a, facilitates Vegfb transcription, by which YBX1 knockdown consequently suppressed the increase of Vegfb mRNA induced by nerve injury. Meanwhile, our results also suggested another YBX1 independent mechanism for Vegfb upregulation, in that circAnks1a sponges constitutive miR-324-3p-mediated suppression of Vegfb mRNA. In another word, in the setting of loss of YBX1 function (knockdown by siRNA), nerve injury might moderately increase VEGFB protein expression without change of Vegfb mRNA, which implying that additional mechanism exists to enhance the translational availability and efficacy of constitutive mRNA in dorsal horn following nerve injury. The present study found that upregulation of circAnks1a may sponge the constitutive miR-324-3p, which consequently facilitates the translation of Vegfb mRNA and upregulates VEGFB expression in the rodents with nerve injury. We hope this could likely address the differential expression of Vegfb mRNA and protein after YBX1 knockdown.

Comments 10: Line 86, “by criteria of at least one unique back-spliced reads” does not make sense thus needs to be reworded. - Line 93, perhaps change “unregulated” to “dysregulated”. - Line 402, “translational” should be replaced by “posttranscriptional”. - SNL acronym is in the title and not defined till the methods.

Responses: According to the reviewer’s suggestion, we have corrected all language error mentioned by the reviewer in the revised manuscript.

Reviewer #3

Comments 1: The authors profiled the abundance of circRNA based on ribo-RNA-seq. It is not clear what computational tool they used for identifying circRNA and estimating their abundance. They claimed only one unique junction read was required as evidence of circRNA expression. Did they estimate the false positive rate based on such low criteria? Also it is not clear whether they required the back-splicing junctions cover the known splicing sites. If so, how can some circRNAs derive solely from intronic sequences? If not, what would be the biogenesis mechanisms for those not covering splicing sites.

Responses: We adopted the computational pipeline previously described (Zhang, et al. *Cell* 2014; 159: 134-147) to identify spliced back-spliced junctions and find circRNAs¹⁸. Briefly, the rRNA-removed reads were first mapped to rat reference genome (Rnor_6.0) by TopHat2, which is a fast splice junction mapper for RNA-Seq reads²¹. After aligning with the reference genome, the unmapped reads were then remapped using TopHat-fusion module to identify back-spliced junction reads²². Back-spliced junction reads were further realigned against the existing gene annotations, and those with 1-2 nucleotide shifted alignments against canonical splice sites were adjusted. Reads with alignments on non-canonical splice sites were discarded. The identified circRNA was called if it was supported by at least one unique back spliced read at least in one sample. We next used HTSeq to obtain the count values of each circRNA and further calculated the Reads Per Million mapped reads (RPM) to estimate their abundance²³.

We set one unique back-spliced junction read threshold, consistent with the previous studies which also set threshold to only 1-2 junction reads^{11,18}. The screening criteria (RPM > 0.1) of circRNAs benefit the reduction of false positive. While we recognized the existence of false positive rate in the sequencing results, the expression of circRNAs (including circAnks1a) was further validated by real-time PCR assay. Actually, in the sequencing results, the targeted circAnks1a had at least 76 junction reads in any single sample. We further showed that manipulation of circAnks1a substantially regulated pain behavior in the rats with nerve injury.

In the process of identifying circRNA, we set the requirement that the back-spliced junctions should cover the known splicing sites. When realigning against existing gene annotations, back-spliced junctions with 1-2 nucleotide shifted alignments against canonical splice sites were adjusted to the correct positions, and those with

non-canonical splice sites were discarded. We also found that some back-spliced junction reads could align to intronic or other regions. If they met the above requirements and also covered the known splicing sites, it would be identified as circRNAs.

Comments 2: The whole study was focused on one circRNA, circAnks1a. It is therefore important to validate its circular feature and its full-length sequences. The golden standard for circRNA validation is Northern Blot, with and without RNase R treatment, with and without linearization.

Responses: According to the reviewer's suggestion, we performed the Northern Blot experiment. The results showed that circAnks1a can be detected with and without RNase R using circAnks1a probe. When detecting with mAnks1a probe, the mAnks1a only can be detected without RNase R (Comment Fig. 22). These results are the validation of circAnks1a. We have added these results into Fig. 2e in the revised manuscript.

Comment Fig. 22

Comments 3: The authors did not show the specificity of siRNA knockdown. Did it also affect linear mRNA?

Responses: To achieve specificity in circRNA-knockdown, all these nucleases must be guided selectively to the circRNA-specific backsplice-junction²⁴. In fact, we designed the circAnks1a siRNA according to this rule as shown in the below Figure (Comment Fig. 23A), and observed the specificity of circAnks1a siRNA knockdown in the present study. To further confirm the result, we synthesized the circAnks1a siRNA and mAnks1a siRNA, and validated their efficacy to reduce circAnks1a or mAnks1a, respectively. The results showed that application of circAnks1a siRNA, but not mAnks1a siRNA, significantly decreased the expression of circAnks1a in naïve

animals (Comment Fig. 23B). We added these contents into Supplementary Fig. 1c and 1d in the revised manuscript.

Comment Fig. 23

Comments 4: The authors did not describe the design of circRNA overexpression constructs and estimated the efficiency of its circularization.

Responses: We appreciated the great comments. The vector was constructed mainly according to Dongming Liang's method to facilitate circularization of circRNA¹⁴. We constructed the full-length sequence of circAnks1a into a modified vector pAAV-CMV-polyA. Importantly, a pair of short inverted repeats (CR1 and CR1 RC) was inserted flanking the full-length sequence of circAnks1a. The short inverted repeats (36nt) will base-pair to one another, thereby bringing the splice sites into close proximity to each other. The sequences of CR1 and CR1 RC were cgctgtcggataatgtgggcacagccgagccgtgtt and aacacggctcggctgtgcccacattatccgacagcg, respectively. Next, we transfected the plasmid into 293T cells and collected cellular RNA after 48 hours. The relative high expression of circAnks1a in qRT-PCR using divergent primers of circAnks1a suggested a high efficiency of circularization (Comment Fig. 24).

Comment Fig. 24

Comments 5: The FISH experiments were done without enough control. The authors should use probes targeting at the sequences shared by linear and circular Anks1a, as well as those targeting only at linear Anks1a, as control.

Responses: According to the reviewer's suggestion, we performed the FISH experiments using the probes targeting at the sequence shared by linear and circular Anks1a (both Anks1a), linear Anks1a (mAnks1a) and circAnks1a. We found that the immunoreactivity with circAnks1a probes and both Anks1a probes, but not mAnks1a probes, was significantly increased in the dorsal horn in the rats with nerve injury (Comment Fig. 25).

Comment Fig. 25

Comments 6: The authors claimed that circAnks1a could enhance the binding between YBX1 and transportin-1. Whereas they showed the binding of YBX1 and circAnks1a, they did not test the interaction between circAnks1a and transportin-1. They also should identify the exact binding sites of both YBX1 and transportin-1, and then compare the effect between overexpression of wildtype circAnks1a and circAnks1a with the binding sites mutated.

Responses: According to the reviewer's suggestion, two potential YBX1 motifs were observed in the circAnks1a based on *in silico* analysis (Comment Fig. 26A). The further EMSA experiment showed that recombinant YBX1 protein can obviously bind to the circAnks1a site-1 (YBX1 motif-1), but not site-2 (YBX1 motif-2) (Comment Fig. 26B), and preincubation of unlabeled probe, but not mut-unlabeled probe, prevented the binding of YBX1 and circAnks1a site-1 (Comment Fig. 26B). These results suggested that the exact binding site of circAnks1a for YBX1 is UCCAGCAA sequence. Next, we verified the interaction between circAnks1a and transportin-1, and found that SNL significantly increased the level of circAnks1a immunoprecipitated by transportin-1 antibody by using RIP methods, which was suppressed by YBX1 siRNA (Comment Fig. 26C). Further EMSA showed that recombinant transportin-1 protein cannot bind to the circAnks1a site 1 (Comment Fig. 26D). These results suggested that transportin-1 did not directly interact with circAnks1a, and the interaction between circAnks1a and transportin-1 was potentially bridged by YBX-1. Furthermore, we observed the effect of overexpression of wildtype circAnks1a and circAnks1a with the binding sites mutated in 293T cells, and found that the binding between YBX1 and mutated circAnks1a was significantly decreased relative to that of wildtype circAnks1a (Comment Fig. 26E). These results were added into the Fig.6 and Supplementary Fig. 6 in the revised manuscript.

Comment Fig. 26

Comments 7: The authors claimed that circAnks1a could enhance the binding of YBX1 at VEGFB promoter. However, they did not provide the direct evidence, i.e. compare the binding of YBX1 at VEGFB promoter with to without circAnks1a overexpression by ChIP.

Responses: According to the reviewer's suggestion, we performed ChIP assay in naïve

rats with circAnks1a overexpression. The results showed that the recruitment of YBX1 in the Vegfb promoter was increased in circAnks1a overexpression (21 days after injection of AAV2/9-circAnks1a-EGFP), when compared to AAV2/9-EGFP (Comment Fig. 27). We have added these results into Supplementary Fig. 7 in the revised manuscript.

Comment Fig. 27

Comments 8: The authors claimed that circAnks1a could affect VEGFB translation via sponging the repressive mir-324-3p. One essential control experiment would be to compare the effect between overexpression of wildtype circAnks1a and circAnks1a with the potential mir-324-3p binding sites mutated.

Responses: According to the reviewer's suggestion, we have added another control experiment to compare the effect between overexpression of wildtype circAnks1a and circAnks1a with the potential miR-324-3p binding sites mutated. The results showed that circAnks1a overexpression rescued the decreased luciferase signal by miR-324-3p while overexpression of circAnks1a with the mutated miR-324-3p binding sites did not rescue the decrease (Comment Fig. 28). We have added these results in Fig. 10I in the revised manuscript.

Comment Fig. 28

Comments 9: Linear Anks1a mRNA covered all the circAnks1a sequence except the junction. Could linear Anks1a have the same effect? The authors should also estimate the relative abundance between its linear and circular isoforms in different conditions.

Responses: We are sorry that we didn't write clearly in the previous submission.

According to the reviewer's suggestion, we observed the role of the linear Anks1a (mAnks1a) in different conditions and found that mAnks1a siRNA can neither change the pain threshold of normal rats (Comment Fig. 29A) nor alleviated the mechanical allodynia (Comment Fig. 29B) induced by SNL.

Furthermore, we examined the relative amounts of circAnks1a in the dorsal horn tissue compared to the linear Anks1a mRNA in the naïve group, sham group and SNL group. We found that the circAnks1a expression has no difference between the naïve group and sham group, while SNL significantly increased the level of circAnks1a RNA, but not linear Anks1a mRNA in the dorsal horn (Comment Fig. 29C). The expression of linear mAnks1a mRNA has no difference among naïve, sham and SNL groups. Furthermore, the ratio between circAnks1a/mAnks1a was significantly increased in SNL rats compared with sham rats (Comment Fig. 29D).

Comment Fig. 29

Comments 10: Finally, the authors have applied quite a few technically challenging methods, such as ChIRP, RNA pull-down assays, and etc. However, there is no detailed description in the Method part. It is therefore impossible to judge the quality of their data.

Responses: According to the peer's suggestion, we added the detailed information about ChIRP and RNA pull-down assay, and other technical details as well, in the revised manuscript.

1. Koch S, Claesson-Welsh L. Signal transduction by vascular endothelial growth factor receptors. *Cold Spring Harb Perspect Med* **2**, a006502 (2012).
2. Lal N, Puri K, Rodrigues B. Vascular Endothelial Growth Factor B and Its Signaling. *Frontiers in cardiovascular medicine* **5**, 39 (2018).
3. Todd AJ, Puskar Z, Spike RC, Hughes C, Watt C, Forrest L. Projection neurons in lamina I of rat spinal cord with the neurokinin 1 receptor are selectively innervated by substance p-containing afferents and respond to noxious stimulation. *The Journal of neuroscience : the official journal of the Society for Neuroscience* **22**, 4103-4113 (2002).
4. Todd AJ. Neuronal circuitry for pain processing in the dorsal horn. *Nat Rev Neurosci* **11**, 823-836 (2010).
5. Deng SJ, *et al.* Hypoxia-induced LncRNA-BX111 promotes metastasis and progression of pancreatic cancer through regulating ZEB1 transcription. *Oncogene*, (2018).
6. Hubisz MJ, Pollard KS, Siepel A. PHAST and RPHAST: phylogenetic analysis with space/time models. *Brief Bioinform* **12**, 41-51 (2011).
7. Memczak S, *et al.* Circular RNAs are a large class of animal RNAs with regulatory potency. *Nature* **495**, 333-338 (2013).
8. Kent WJ, *et al.* The human genome browser at UCSC. *Genome Res* **12**, 996-1006 (2002).
9. Huang S, *et al.* Loss of Super-Enhancer-Regulated CircRNA Nfix Induces Cardiac Regeneration After Myocardial Infarction in Adult Mice. *Circulation*, (2019).
10. Dragomir M, Calin GA. Circular RNAs in Cancer - Lessons Learned From microRNAs. *Front Oncol* **8**, 179 (2018).
11. Zheng Q, *et al.* Circular RNA profiling reveals an abundant circHIPK3 that regulates cell growth by sponging multiple miRNAs. *Nature communications* **7**, 11215 (2016).
12. Peng S, *et al.* Circular RNA SNX29 Sponges miR-744 to Regulate Proliferation and Differentiation of Myoblasts by Activating the Wnt5a/Ca(2+) Signaling Pathway. *Mol Ther Nucleic Acids* **16**, 481-493 (2019).
13. Smid M, *et al.* The circular RNome of primary breast cancer. *Genome Res* **29**, 356-366 (2019).
14. Liang D, Wilusz JE. Short intronic repeat sequences facilitate circular RNA production. *Genes Dev* **28**, 2233-2247 (2014).

15. Yu J, *et al.* Circular RNA cSMARCA5 inhibits growth and metastasis in hepatocellular carcinoma. *Journal of hepatology*, (2018).
16. Yang Y, *et al.* Novel Role of FBXW7 Circular RNA in Repressing Glioma Tumorigenesis. *Journal of the National Cancer Institute* **110**, (2018).
17. Wei X, *et al.* Circular RNA profiling reveals an abundant circLMO7 that regulates myoblasts differentiation and survival by sponging miR-378a-3p. *Cell death & disease* **8**, e3153 (2017).
18. Zhang XO, Wang HB, Zhang Y, Lu X, Chen LL, Yang L. Complementary sequence-mediated exon circularization. *Cell* **159**, 134-147 (2014).
19. Tong J, Sydorsky Y, St-Germain JR, Taylor P, Tsao MS, Moran MF. Odin (ANKS1A) modulates EGF receptor recycling and stability. *PloS one* **8**, e64817 (2013).
20. Kim J, Lee H, Kim Y, Yoo S, Park E, Park S. The SAM domains of Anks family proteins are critically involved in modulating the degradation of EphA receptors. *Mol Cell Biol* **30**, 1582-1592 (2010).
21. Kim D, Pertea G, Trapnell C, Pimentel H, Kelley R, Salzberg SL. TopHat2: accurate alignment of transcriptomes in the presence of insertions, deletions and gene fusions. *Genome biology* **14**, R36 (2013).
22. Kim D, Salzberg SL. TopHat-Fusion: an algorithm for discovery of novel fusion transcripts. *Genome biology* **12**, R72 (2011).
23. Anders S, Pyl PT, Huber W. HTSeq--a Python framework to work with high-throughput sequencing data. *Bioinformatics* **31**, 166-169 (2015).
24. Holdt LM, Kohlmaier A, Teupser D. Circular RNAs as Therapeutic Agents and Targets. *Front Physiol* **9**, (2018).

REVIEWERS' COMMENTS:

Reviewer #1 (Remarks to the Author):

The authors have collected additional data to address concerns that I previously raised. They responded to almost all issues in the revised manuscript.

1) In response to my Comment 3, the authors showed additional electrophysiological data. In the left panel of Supplementary Figure 2a and 2b, please check the values of scale bars. Is 10 mV right? Also please check the values of injection currents in the abscissa in the graph in Figure 2a. Although the authors do not show the input resistance of these cells, 20 pA-current injection could cause only 10 mV-depolarization at best (much lower than the representative figures) in the cells with 500 M Ohm (an assumed value from previous studies).

The figure legend of Supplementary Figure 2a (line 1) seems to be depolarizing current injection-induced action potential, not C-fiber-evoked one.

2) In response to Comment 15, the authors clearly described methods in more detail. As regards RNA pulldown assay, the references seem to be incorrectly cited (line 2). Please check them.

Reviewer #2 (Remarks to the Author):

I have no further comments.

Reviewer #3 (Remarks to the Author):

All my comments have been addressed.

REVIEWERS' COMMENTS:

Reviewer #1

Comment 1: In response to my Comment 3, the authors showed additional electrophysiological data. In the left panel of Supplementary Figure 2a and 2b, please check the values of scale bars. Is 10 mV right? Also please check the values of injection currents in the abscissa in the graph in Figure 2a. Although the authors do not show the input resistance of these cells, 20 pA-current injection could cause only 10 mV-depolarization at best (much lower than the representative figures) in the cells with 500 M Ohm (an assumed value from previous studies).

The figure legend of Supplementary Figure 2a (line 1) seems to be depolarizing current injection-induced action potential, not C-fiber-evoked one.

Response: We appreciate the great comments from the reviewer. Actually there existed an error in scale bar in Supplementary Figure 2a and 2c, as pointed out by the reviewer. We rechecked the original data and made appropriate correction on the scales in the current submission.

We also corrected “C-fiber-evoked action potentials” into “depolarizing current injection-induced action potentials” in Supplementary Figure 2.

Comment 2: In response to Comment 15, the authors clearly described methods in more detail. As regards RNA pulldown assay, the references seem to be incorrectly cited (line 2). Please check them.

Response: Thank you very much for your careful review. According to the editor and your advice, we have revised the references in the method of RNA pulldown assay.